# The potential bidirectional relationship between long COVID and menstruation

Jacqueline A. Maybin [1] ✉, Catherine Walker[1], Marianne Watters [1], Natalie ZM. Homer [2], Joanna P. Simpson [2], Cara Robb[1], Douglas A. Gibson [1], Luna Jeanjean[3], Hilary O. D. Critchley [1], Gabriella Kountourides[4], Zuzanna Olszewska[4] & Alexandra Alvergne [3,4]

Women have reported menstrual changes following SARS-CoV-2 infection and variation in long COVID symptoms across the menstrual cycle. We examined (i) whether COVID is linked to abnormal uterine bleeding (AUB), (ii) if long COVID symptoms vary with the menstrual cycle, and (iii) potential underlying mechanisms. Here we show long COVID was associated with AUB in a UK population. When compared to those never infected (n = 9423), long COVID participants (n = 1048) reported increased menstrual volume, duration and intermenstrual bleeding, while those who recovered from acute COVID (n = 1,716) reported minimal menstrual disruption. Long COVID symptoms examined in 54 women across the menstrual cycle revealed that severity was highest during the perimenstrual and proliferative phases. Serum and endometrial analysis revealed higher serum 5α-dihydrotestosterone and lower endometrial androgen receptors in long COVID versus no COVID. Other ovarian hormones showed no significant differences. Serum cytokine profiling indicated increased menstrual inflammation with long COVID and immune cell aggregates were observed in menstrual endometrium. In conclusion, long COVID was associated with AUB but not impaired ovarian function. Differences in peripheral and endometrial inflammation may contribute to AUB and long COVID symptom severity. We anticipate our findings will instigate exploration of new therapeutic strategies for women with long COVID.

There have been many reports of menstrual disturbance from women and people who menstruate who are suffering from long-term symptoms of COVID-19. Menstrual bleeding occurs following the decline in ovarian sex hormone production at the end of the menstrual cycle, resulting in the shedding of the upper two-thirds of the endometrium at menstruation. Menstrual symptoms have been standardised and defined by the International Federation of Gynecology and Obstetrics (FIGO) AUB System 1[1]. Menstrual bleeding typically has a frequency of every 24–38 days, duration of no more than 8 days, variation in shortest to longest menstrual cycle of less than or equal to 7–9 days

(age dependent) and a flow volume of subjectively normal. Common AUB symptoms are frequent/infrequent, prolonged, irregular and heavy menstrual bleeding (HMB). The symptom of HMB is defined as excessive menstrual blood loss that interferes with a woman's physical, emotional, social and/or material quality of life[2].

Pre-pandemic, AUB was extremely prevalent. Depending on the population studied and definition adopted, global figures vary but were consistently high[3,4]. One in three women was reported to find their menstrual loss excessive, with this figure rising to one in two as the menopause approaches[5,6]. Over 800,000 women sought

[1]Centre for Reproductive Health, University of Edinburgh, Edinburgh, UK. [2]Mass Spectrometry Core, Edinburgh Clinical Research Facility, Centre for Cardiovascular Sciences, University of Edinburgh, Edinburgh, UK. [3]ISEM, University of Montpellier, CNRS, IRD, EPHE, Montpellier, France. [4]School of Anthropology and Museum Ethnography, University of Oxford, Oxford, UK. ✉e-mail: jackie.maybin@ed.ac.uk

treatment for HMB per year in the UK alone, with many more suffering in silence[3,7]. These menstrual symptoms can have a profoundly negative impact on quality of life, interfering with physical, social, mental and material wellbeing[2,8]. HMB is a leading cause of iron deficiency anemia in developed countries[9], and, when extreme, can necessitate blood transfusion. HMB also affects work productivity, with results from a US study demonstrating that those with self-reported HMB were less likely to be working[10]. The annual indirect cost of menstrual bleeding disorders in the US was estimated to be $12 billion[11]. Therefore, any increase in prevalence of AUB due to COVID-19 has the potential to increase the gender health gap and add to the financial burden for health services and the economy.

Menstrual disturbances reported during the COVID pandemic may be due to COVID-19 vaccination, infection with the SARS-CoV-2 virus or pandemic-related stress and/or lifestyle changes. The contribution of each factor to menstrual disturbance is beginning to be delineated[12–14] but is hindered by the lack of menstrual data collected during the pandemic using standardized nomenclature[1] to facilitate scientific and clinical comparison of menstrual data globally. When examining COVID-19 vaccination and AUB, studies have revealed small changes in menstrual frequency[15–18] but were consistent in their findings that any menstrual disturbance related to vaccination was transient, similar to findings with other vaccines[16,19–21]. Our previous UK-based online survey[22] revealed that 18% of women reported a change to their menstrual symptoms after vaccination, but that menstrual symptoms were not significantly different in the COVID vaccination group when compared to those who had not been vaccinated. The impact of acute COVID-19 on menstruation was also examined by a few small studies in China early in the pandemic, revealing an association with menstrual disturbance[23,24]. These associations were found in subsequent larger, community-based US and UK studies[22,25], consistent with infection with SARS-CoV-2 having a larger effect on menstruation than vaccination.

AUB and reproductive health in those with sustained post-COVID-19 infection sequelae (long COVID or long haul COVID) has been significantly understudied[26]. A recent WHO-led Delphi process reached a consensus that this post-COVID-19 condition occurs in individuals with a history of probable or confirmed SARS-CoV-2 infection, usually 3 months from the onset, with symptoms that last for at least 2 months and cannot be explained by an alternative diagnosis[27]. A patient-led survey of those experiencing long COVID symptoms revealed nine main symptom clusters: systemic, reproductive/genitourinary/endocrine, cardiovascular, musculoskeletal, immunologic/autoimmune, head, eyes, ear, nose and throat, pulmonary, gastrointestinal and dermatologic[28]. Evidence suggests that long COVID affects twice as many women as men and, of those under 50, women were five times less likely to report feeling recovered than men of the same age[29]. A survey of those with long COVID revealed menstrual issues were reported by 33.8%[28]. In a Spanish survey, those with suspected/diagnosed long-COVID-19 (n = 748) had an increased risk of self-reported menstrual alterations when compared to those who had never had COVID-19 or those with acute COVID-19 who had recovered[30]. These initial data suggest long COVID is associated with menstrual disturbance, but the type of menstrual disturbance was not reported.

As well as long COVID affecting menstrual symptoms, the ovarian hormone fluctuations across the menstrual cycle have the potential to impact the symptoms of long COVID. A cross-sectional study of long COVID patients found that over one-third of menstruating patients reported an exacerbation of their symptoms the week before or during menses[28]. The menstrual cycle and ovarian sex hormones may also modulate COVID disease susceptibility and severity[26,31], with oestradiol known to boost the host immune response[32]. Women under 50 years old were reported to have an increased risk of developing long COVID[29], consistent with ovarian sex hormones increasing the likelihood of developing long COVID. However, the effects of the menstrual cycle on longer-term symptoms of COVID-19 remain under-researched.

Defining the mechanisms that underpin any AUB associated with long COVID will inform clinical management. The underlying cause(s) of AUB may be classified as structural or non-structural, as outlined in the FIGO AUB System 2[1]. Structural causes are those that can usually be detected during routine examination or investigations (e.g., imaging, histopathology) and include Polyps, Adenomyosis, Leiomyomas (fibroids) or Malignancy (PALM). Non-structural causes are not detected on imaging and include Coagulopathies, Ovulatory disorders, primary Endometrial disorders, Iatrogenic causes and those that are Not otherwise classified (COEIN). The rapid onset of menstrual disturbance described with SARS-CoV-2 infection favours a non-structural cause such as an ovulatory or endometrial disorder. Development and shedding of the endometrium at menstruation is controlled by the ovarian hormones. Perturbations of the cyclical ovarian sex hormone production (e.g., ovulatory disorders) can lead to changes in menstrual regularity and volume. Those with AUB due to endometrial disorders have previously been shown to have excessive endometrial inflammation at the time of menstruation[33,34]. Whether similar aberrations are present in those with long COVID remains to be determined. Defining the underlying mechanisms of AUB associated with long COVID will facilitate the precise treatment of menstrual disturbance. Similarly, uncovering the mechanisms that result in increased severity or number of long COVID symptoms across the menstrual cycle may reveal new treatment options for females suffering from long COVID.

Hence, we hypothesised (i) that long COVID is associated with increased reports of AUB, (ii) women with long COVID experience increased number and severity of their long COVID symptoms prior to and during menstruation, (iii) that those with long COVID have altered ovarian sex hormones production or response and/or excessive peripheral or endometrial inflammation. We tested these hypotheses using three approaches: (i) a large online UK COVID and reproductive health survey, (ii) a longitudinal study of long COVID symptoms across the menstrual cycle, and (iii) collection and analysis of carefully categorised biological samples of serum and endometrium at three phases of the menstrual cycle from those with and without long COVID.

## Results

### In a UK survey, menstrual volume, duration and intermenstrual bleeding were increased in those with long COVID when compared to controls

**Sample characteristics.** Out of the 26710 individuals who completed our online survey, "The COVID-19 Pandemic and Women's Reproductive Health", we excluded participants who did not have a menstrual bleed in the 12 months preceding the survey, those who were post-menopausal or peri-menopausal, breastfeeding or pregnant, those who did not live in the UK, those enrolled in a clinical trial, and those with unknown vaccine, COVID or long COVID status. Since Long COVID was defined as infection ≥30 days ago, we excluded participants with very recent acute COVID infections (in the last 30 days) to ensure temporal comparability and prevent overrepresentation of newer variants in the acute COVID group (See Suppl. Fig. 1 for variant distribution across groups). The final sample size was 12187, of which 9423 (77%) had never been diagnosed with COVID (no COVID group), 1716 (14%) of participants reported previous acute COVID and 1048 (9%) had long COVID (Table 1). 4814 (40%) reported having been vaccinated, with either one (n = 4009) or two doses (n = 832). The median age was 36 years old (inter-quartile range (IQR): 29–43) for those with long COVID, while it was 31 years old (inter-quartile range (IQR): 25–39) for those with previous acute COVID and 32 years old (inter-quartile range (IQR): 25–40) for those who did not report COVID. Of note, 92% of participants were white, 64% were nulliparous, and 47% had a university or college degree. 57% reported one or more abnormal menstrual symptoms (e.g., irregular cycles, heavy bleeding, frequent

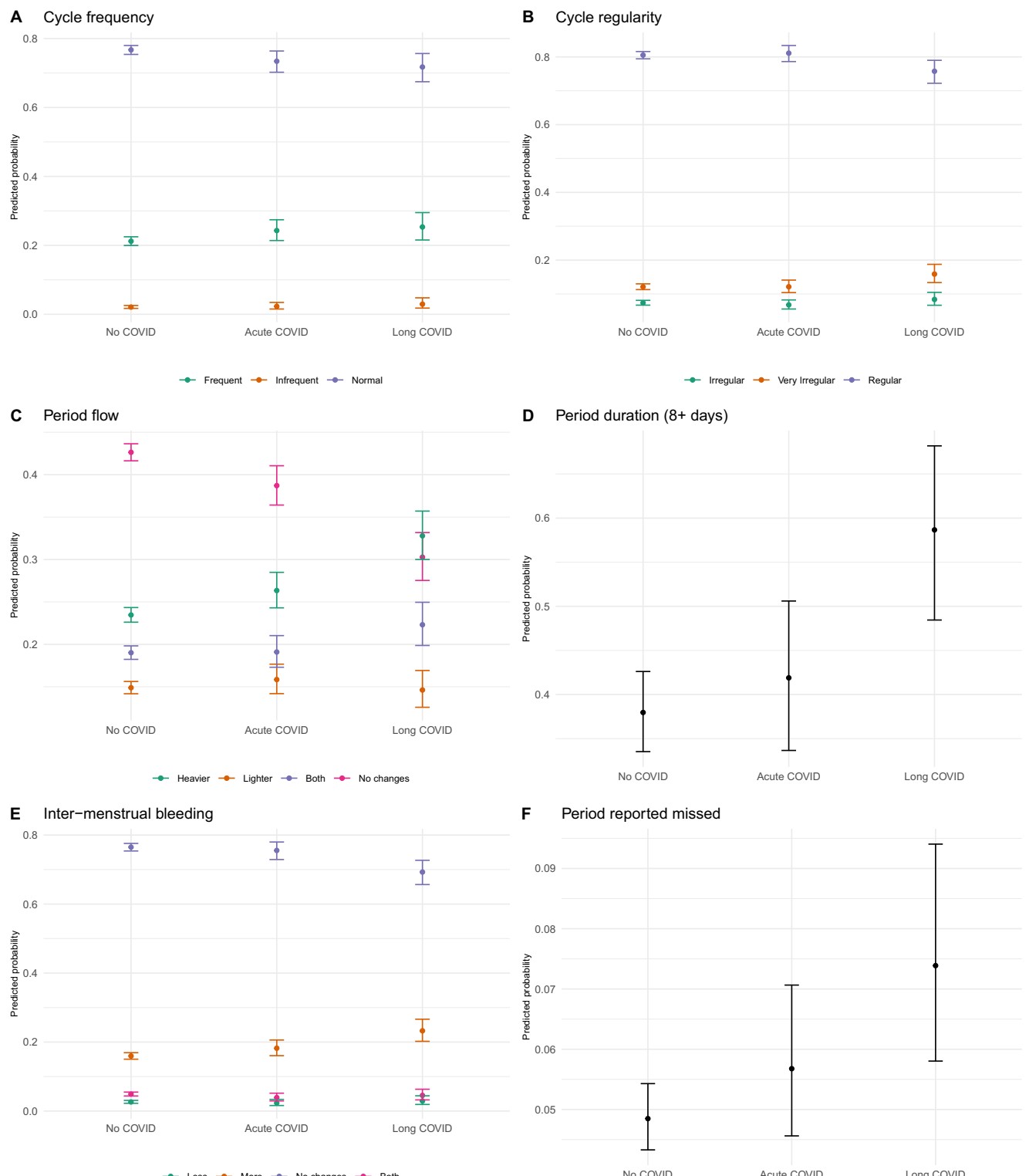

or infrequent cycle length, prolonged periods) at baseline and 20% had a formal diagnosis of a condition known to affect reproductive function (e.g., endometriosis, polycystic ovary syndrome, HIV/AIDS, underactive/overactive thyroid, uterine polyps, uterine fibroids, eating disorders, interstitial cystitis).

**Risk for "abnormal" menstrual characteristics.** To investigate the independent effects of long COVID on abnormal menstrual parameters as defined by the FIGO criteria for abnormal uterine bleeding, we compared menstrual cycle parameters across 3 groups (Table 1)[1]: No

COVID[2], previous acute COVID and[3] long COVID. The relationships between cycle parameters and the history of COVID-19 disease were adjusted for relevant menstrual cycle parameters before the pandemic, age, BMI, exogenous hormone use, and presence of diagnosed reproductive condition at baseline.

*Menstrual Frequency (n = 9843, including 817 Long COVID, 1382 Acute COVID and 7644 controls).* Data for menstrual frequency were missing for 1455 participants, and these were excluded from this specific analysis. We also excluded participants who reported "Too irregular to say" for the outcome variable "Cycle length during the

**Fig. 1 | Reported menstrual symptoms in a UK population of people who menstruate and (1) have never had COVID-19 (No COVID), (2) have had acute COVID-19 and recovered (Acute COVID), or (3) experienced long-term symptoms of COVID-19 by the time of the survey (Long COVID). A** Menstrual cycle frequency ($n = 9843$): Participants reported cycles <24 days (green), 24–38 days (blue), or >38 days (orange). No significant association was found between disease group and abnormal (frequent or infrequent) versus normal cycle frequency (FDR-adjusted $p = 0.1296$ for Long COVID). **B** Menstrual cycle regularity ($n = 12,187$): participants reported regular cycles with <10 days difference between shortest and longest cycles (blue), somewhat irregular cycles with 10–20 days difference (green), or highly irregular cycles with >20 days difference (orange). No significant association was found between disease group and irregular versus regular cycles (FDR-adjusted $p = 0.0909$ for Long COVID). **C** Subjective menstrual flow volume ($n = 12,187$): Participants reported no changes (pink), heavier flow (green), lighter flow (orange), or both lighter and heavier flow (blue). Compared to participants with no COVID history, the relative risk of reporting heavier versus normal flow was significantly elevated in those with long COVID (RR = 1.93, 95% CI: 1.59–2.35, FDR-adjusted $p = 0.00000000346$) and marginally elevated in those with acute COVID

(RR = 1.19, 95% CI: 1.03–1.38, FDR-adjusted $p = 0.0538$). **D** Menstrual duration ($n = 1938$): participants reporting periods >8 days. Compared to those with no COVID history, the prevalence of extended menstrual duration was significantly higher among participants with Long COVID (PR = 2.26, $p = 0.0006$) but not among those with acute COVID (PR = 1.22, $p = 0.3332$). **E** Intermenstrual bleeding ($n = 12,187$): Participants reported less than usual bleeding (green), more than usual bleeding (orange), no change (blue), or both less and more bleeding (pink). Compared to participants with no COVID history, the relative risk of reporting increased intermenstrual bleeding versus no changes was significantly elevated in those with long COVID (RR = 1.59, 95% CI: 1.29–1.97, FDR-adjusted $p = 0.0001$) but not in those with acute COVID (RR = 1.15, 95% CI: 0.97–1.36, FDR-adjusted $p = 0.22$). **F** Missed menstrual episodes ($n = 12,187$): Compared to participants with no Covid history, the prevalence of missed periods was significantly higher among those with long COVID (PR = 1.39, $p = 0.0033$) but not among those with acute COVID (PR = 1.15, $p = 0.17$). Graphs show mean predicted probabilities and 95% confidence intervals. Data and code are available on GitHub https://github.com/ataquette/Long-COVID-Mens.

---

pandemic" ($n = 889$), as we were interested in ascribing frequency. Another 1445 individuals were excluded due to missing data for baseline menstrual frequency before the pandemic. Across all groups of remaining participants, the most probable outcome was to report normal cycles (between 24 and 38 days, 71%), followed by frequent (<24 days, 26%) and infrequent cycles (>38 days, 3%) (Fig. 1A). The relative risk or risk ratio (RR) of reporting abnormal (frequent vs. normal cycles or infrequent vs. normal cycles) does not vary across groups, Table S1).

*Menstrual regularity ($n = 12,187$).* Across all groups of participants, the most probable outcome is to report regular cycles at the time of survey (less than 10 days difference between shortest and longest cycles, 79.7%), followed by very irregular (over 20 days difference, 9.8%), and somewhat irregular (between 10–20 days difference, 10.5%, Fig. 1B). The risk of reporting somewhat irregular vs. regular cycles increased by 39% for those with Long COVID, although the difference is not significant at the false discovery rate (RRR = 1.39, 95% CI = [1.03 to 1.87], $p$ value = 0.04, FDR $p$ value = 0.09; Table S2). The relative risks of reporting irregular vs. regular cycles are associated with obesity and the use of exogenous hormones (increased with progestin-based therapies and decreased with combined contraceptives).

*Flow volume ($n = 12,187$).* Across all groups of participants, the most probable participant reported outcome was "no changes" (41.0%), followed by "heavier" (24.9%), "heavier, and lighter" (19.2%) and "lighter" (14.9%) (Fig. 1C). As compared to control, a history of COVID-19 disease increased the risk of "heavier" vs. "normal" flow by ca. 19% for previous acute COVID (RRR = 1.19, 95% CI = [1.03 to 1.38], FDR $p$ value = 0.05) and by 93% for long COVID (RR = 1.93, 95% CI = [1.59 to 2.35], FDR $p$ value < 0.001). The risk of "lighter" flow vs. "no changes" increased by 36% for long COVID (RRR = 1.36, 95% CI = [1.08 to 1.71], FDR $p$ value = 0.03) but did not vary between the control group and the acute COVID group. Long COVID also increased the risk of "lighter and heavier" flow as compared to "no changes" by 57% (RRR = 1.57, 95% CI = [1.26 to 1.96], FDR $p$ value < 0.001, Table S3).

*Menstrual duration ($n = 1,938$ including 192 Long COVID, 288 Acute COVID and 1458 controls).* Data for menstrual duration were missing for 10,249 participants, and these were excluded from this specific analysis. Compared to the control group, the prevalence of menstruation lasting longer than 8 days is increased twofold for the long COVID group (PR = 2.26, 95% CI [1.46 to 3.49], FDR $p$ value < 0.001), a tendency not observed for those with previous acute COVID-19 disease only (PR = 1.22, 95% CI [0.85; 1.77], FDR $p$ value = 0.33, Table S4, Fig. 1D). The prevalence of periods longer than 8 days decreased with age but increased with abnormal menstrual duration at baseline and presence of diagnosed reproductive pathology at baseline, copper IUD use and the use of progestin-based contraceptives.

*Intermenstrual bleeding or "spotting" ($n = 12,187$).* As compared to the no COVID group, a history of COVID-19 increased the risk of more intermenstrual bleeding by 59% for long COVID (RRR = 1.59, 95% CI = [1.29 to 1.97], FDR $p$ value < 0.001), but not for previous acute COVID (RRR = 1.15, 95% CI = [0.97 to 1.36], FDR p value = 0.22) (Fig. 1E). There was no association between a history of COVID-19 and less intermenstrual bleeding. Both more and less inter-menstrual bleeding increased with the use of hormonal contraceptives, the copper IUD, abnormal menstrual symptoms at baseline and presence of diagnosed reproductive pathology at baseline. (Table S5).

*Missed episodes of menstruation ($n = 12,187$).* As compared to the control group with no covid, the prevalence of reported "missed" and/or "stopped" periods increases by 39% in the long COVID group (PR = 1.39, 95% CI = [1.13 to 1.7], FDR $p$ value = 0.003) but not in the previous acute COVID group (PR = 1.15, 95% CI = [0.97 to 1.37], FDR $p$ value = 0.17, Table S6) (Fig. 1F). The probability of reporting missed or stopped periods decreases with age but increases with obesity, presence of diagnosed reproductive pathology at baseline and the use of exogenous hormones, especially progestin-based contraceptives.

In summary, reported menstrual flow volume, menstrual duration (>8 days), inter-menstrual bleeding and missed episodes of menstruation were significantly increased in those with long COVID versus no COVID. Menstrual frequency and regularity were unchanged. In contrast, in those with previous acute COVID, only menstrual volume was increased, but did not reach statistical significance.

## A prospective study of long COVID symptoms revealed increased severity during the peri-menstrual and proliferative phases of the cycle

**Sample characteristics.** A total of 93 individuals with long COVID provided informed consent for our UK-wide app-based prospective survey to examine 29 common long COVID symptoms across the menstrual cycle. Participants were asked to complete a daily survey to record their menstrual bleeding and the number and severity of their COVID symptoms over 3 months. Participants were at least 18 years old, experienced regular menstrual bleeding (cycle length between 24 and 38 days), were not breastfeeding or pregnant, had previously experienced symptoms consistent with acute COVID-19 or had a positive PCR test for COVID-19, and experienced long COVID symptoms. After excluding participants with inconsistent data ($n = 15$, missing IDs), those using exogenous hormones or the copper IUD ($n = 13$) and removing daily entries with no corresponding menstrual cycle phase ($n = 11$), the final sample included 54 participants (Fig. 2A).

In this sample, 100% of participants identified as women and were white, 69% lived in England, and 61% had a college or university education. Participants were aged 21–50 years (mean age ± sd = 40 ± 6).

**Table 1 | Summary demographics for UK survey participants**

| Characteristic | No COVID, N = 9423 | Acute COVID, N = 1716 | Long COVID, N = 1048 |
|---|---|---|---|
| Age, median (IQR) | 32 (25–40) | 31 (25–39) | 36 (29–43) |
| *Body mass Index*, n (%) | | | |
| Healthy weight | 4161 (44) | 699 (41) | 370 (35) |
| Obese | 2515 (27) | 492 (29) | 372 (35) |
| Overweight | 2578 (27) | 498 (29) | 291 (28) |
| Underweight | 169 (1.8) | 27 (1.6) | 15 (1.4) |
| *Contraceptive use at the time of the survey*, n (%) | | | |
| Combined estrogen–progestin | 1085 (14) | 210 (15) | 79 (9.1) |
| Copper IUD | 432 (5.6) | 82 (5.8) | 47 (5.4) |
| None | 4405 (57) | 787 (56) | 520 (60) |
| Other | 155 (2.0) | 21 (1.5) | 14 (1.6) |
| Progestin only | 1474 (19) | 287 (20) | 193 (22) |
| Sterilization | 171 (2.2) | 28 (2.0) | 19 (2.2) |
| Unknown | 1701 | 301 | 176 |
| *COVID status (diagnosis)*, n (%) | | | |
| Negative | 9423 (100) | 0 (0) | 0 (0) |
| Self-diagnosed+ | 0 (0) | 388 (23) | 162 (15) |
| Tested+ | 0 (0) | 1328 (77) | 886 (85) |
| *Number of vaccination shots*, n (%) | | | |
| Unvaccinated | 5788 (61) | 985 (57) | 573 (55) |
| 1 dose | 3023 (32) | 598 (35) | 388 (37) |
| 2 doses | 612 (6.5) | 133 (7.8) | 87 (8.3) |
| *Vaccine type*, n (%) | | | |
| Oxford-AstraZeneca | 1969 (55) | 377 (52) | 273 (58) |
| Pfizer-BioNTech | 1626 (45) | 348 (48) | 198 (42) |
| Unknown | 5828 | 991 | 577 |
| *Income*, n (%) | | | |
| Less than £13,682 | 1708 (18) | 290 (17) | 138 (13) |
| Between £13,682 and £22,140 | 1233 (13) | 235 (14) | 140 (13) |
| Between £22,140 and £29,254 | 1100 (12) | 223 (13) | 155 (15) |
| Between £29,254 and £39,397 | 1398 (15) | 246 (14) | 152 (15) |
| Between £39,397 and £76,144 | 3191 (34) | 558 (33) | 360 (34) |
| More than £76,144 | 793 (8.4) | 164 (9.6) | 103 (9.8) |
| *Parity (number of deliveries*, n (%)) | | | |
| 0 | 6196 (66) | 1098 (64) | 541 (52) |
| 1 | 1067 (11) | 200 (12) | 149 (14) |
| 2 | 1457 (15) | 269 (16) | 237 (23) |
| 3 | 479 (5.1) | 107 (6.2) | 75 (7.2) |
| 4 | 152 (1.6) | 31 (1.8) | 28 (2.7) |
| 5 | 45 (0.5) | 8 (0.5) | 12 (1.1) |
| 6 | 19 (0.2) | 1 (<0.1) | 4 (0.4) |
| 7+ | 8 (<0.1) | 2 (0.1) | 2 (0.2) |

Half of the participants were overweight or obese, and half reported a healthy BMI. Paid work was reported by 61% of participants, and 48% reported caring responsibilities. With regards to COVID-19, participants had first been infected at least 2 months and sometimes over 2 years before taking the survey (mean ±sd = 17 ± 7 months), and when asked to score their wellbeing relative to that before SARS-CoV-2 infection, participants reported a score of 47% (min = 5%; max = 85%). A total of 93% of participants reported being vaccinated against the disease at the time of the study.

The median number of daily entries over the course of the survey was 33 days (IQR [15;47]), with a minimum of 1 day to a maximum of 60 days. The sample included data points from 1140 days, 149 distinct cycles and 131 distinct phases (54 in the late secretory/menstrual (LS/M) phase; 33 in the proliferative (P) phase; 44 in the secretory (S) phase. See methods for definitions of cycle stage. The median number of cycles contributed by participants was 3 (1 cycle (*n* = 8), 2 cycles (*n* = 11), 3 cycles (*n* = 22), 4 cycles (*n* = 12), 5 cycles (*n* = 1)).

**Number of daily Long COVID symptoms across the menstrual cycle.** First, we analysed a subsample of 32 participants who provided data across all three phases of the menstrual cycle to conduct a within-individual analysis. The results showed no significant difference in the median number of patient-defined long COVID symptoms[28] experienced across the three phases ($\chi^2(2) = 2.60$, *n* = 32, *p* = 0.27, Friedman test, Kendall W = 0.04, Fig. 3A, B). Second, we conducted a minimally age-adjusted multilevel Poisson regression to account for repeated measures of symptoms within individuals and phases. This analysis revealed no significant association between the phase of the menstrual cycle and the number of symptoms experienced on a given day ($\chi^2(2) = 2.58$, *p* = 0.28, *n* = 54). Although being in the peri-menstrual (late secretory to menstrual: LS/M) phase was associated with a 3% increase in the incidence rate of the number of distinct symptoms compared to the reference phase (IRR = 1.03, 95% CI [0.79–1.32], Table S7), this association was not statistically significant (*p* = 0.15). Importantly, 75% of the total variance in the number of distinct symptoms was attributable to differences between participants, indicating a high level of symptom clustering within individuals (ICC = 0.75, Table S7).

**Presence of long COVID symptoms across the menstrual cycle phases.** We found significant variation across symptoms in the extent to which they were experienced in this sample: while some symptoms are reported most days by most individuals (e.g., brain fog, memory issues, post-exertional malaise and fatigue), others appear to be infrequent (e.g., elevated temperature, sore throat, Fig. 3C). We conducted a series of minimally age-adjusted logistic regression models to investigate the association between menstrual cycle phases and the presence of each symptom, accounting for repeated measures within individuals and phases. The LS/M phase was associated with an increased risk of experiencing chills/sweats (OR = 2.30, 95% CI [1.63 to 3.22]) and vision issues (OR = 2.08, 95% CI [1.37 to 3.17]) as compared to the secretory phase. By contrast, the risk of experiencing breathlessness decreases in the proliferative phase (OR = 0.47, 95% CI [0.27 to 0.81]) as compared to the secretory phase. There are no differences across phases for other symptoms (Table S8).

**Severity of daily long COVID symptoms across the menstrual cycle phases.** We then conducted a series of unadjusted cumulative link mixed models to analyse the daily severity score recorded on an ordinal scale ("A little", "Quite a bit", "A lot", "Extremely") while adjusting for variability between individuals. We grouped together "A lot" and "Extremely" because of the low number of responses in the "A lot" category (Fig. 3C, Table S9). As compared to the earlier part of the secretory phase, the late secretory/menstruation phase is associated with more severe dizziness (OR = 1.94, 95% CI = [1.28 to 2.95], FDR-adjusted *p* value = 0.005), fatigue (OR = 1.79, 95% CI = [1.38 to 2.33], FDR-adjusted *p* value = 0.000), post-exertional malaise (OR = 1.45, 95% CI = [1.1 to 1.90], FDR-adjusted *p* value = 0.02), muscles aches (OR = 1.48, 95% CI = [1.06 to 2.053], FDR-adjusted *p* value = 0.04), headache

### A. Recruitment to prospective long COVID symptom study across menstrual cycle.

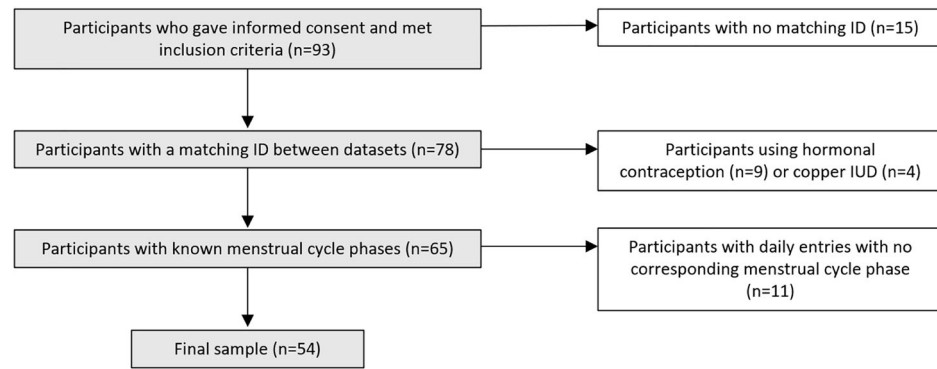

### B. Recruitment to biological studies.

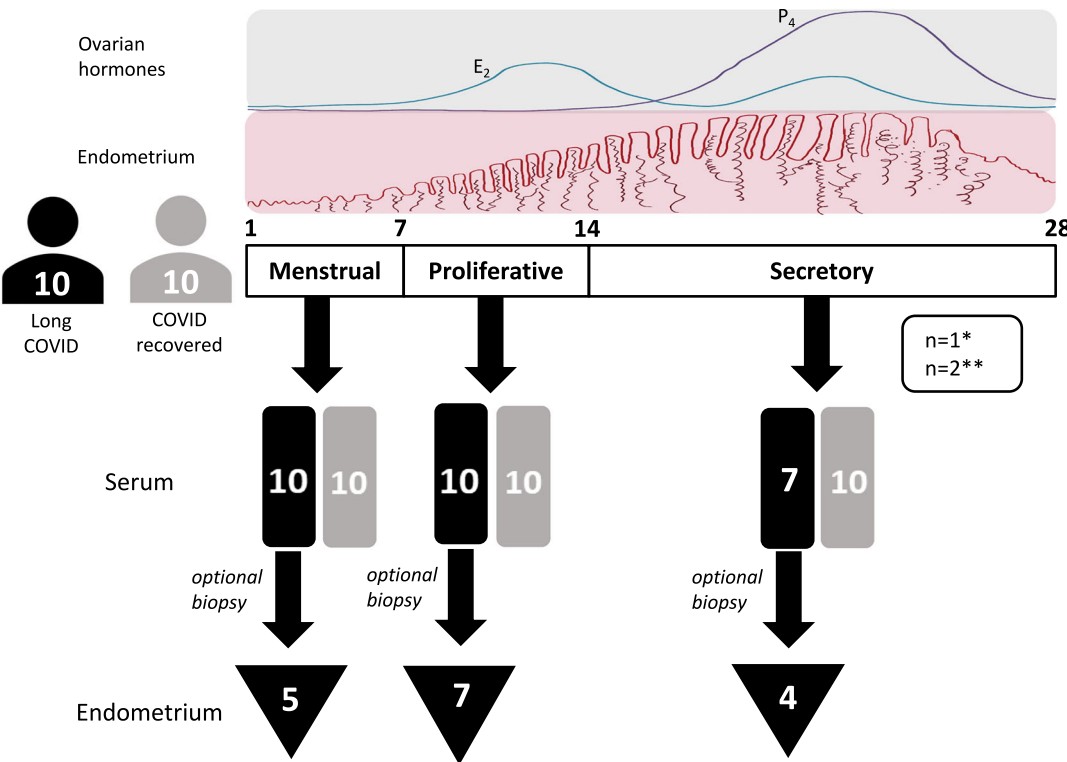

**Fig. 2 | Recruitment of participants to the long COVID symptom study and biological sample study. A** Recruitment to the prospective app-based study of long COVID symptoms across the menstrual cycle (White boxes: exclusions). **B** Recruitment for collection of serum and optional endometrial biopsies from women with long COVID and women who had recovered from acute COVID at three phases of the menstrual cycle. *One long COVID participant became ineligible due to commencing exogenous hormones. ** Two late cycle serum/endometrial samples from women with long COVID were excluded due to anovulatory cycles. $E_2$: estradiol, $P_4$: progesterone.

(OR = 2.106, 95% CI = [1.50 to 2.97], FDR-adjusted $p$ value = 0.000), and tinnitus (OR = 2.35, 95% CI = [1.16 to 4.74], FDR adjusted $p$ value = 0.03). Further, the proliferative phase is associated with more severe post-exertional malaise (OR = 1.63, 95% CI = [1.1 to 2.38], FDR adjusted $p$ value = 0.024), breathing issues (OR = 3.15, 95% CI = [1.36 to 7.26], FDR adjusted $p$ value = 0.015), nausea (OR = 2.25, 95% CI = [1.08 to 4.70], FDR adjusted $p$ value = 0.05) and headache (OR = 2.801, 95% CI = [1.76 to 4.46], FDR adjusted $p$ value = 0.000) as compared to the earlier part of the secretory phase (Fig. 3D).

### Higher serum androgens in the secretory phase and less AR at menstruation may contribute to abnormal uterine bleeding in those with long COVID

Given our findings of an association between long COVID and menstrual disturbance and an increase in COVID symptom severity in the peri-menstrual (LS/M) and proliferative phase, we wished to determine if ovarian sex hormone levels were altered in regularly cycling women experiencing long COVID. Therefore, we collected serum blood samples from 10 women not using exogenous hormones with symptoms of

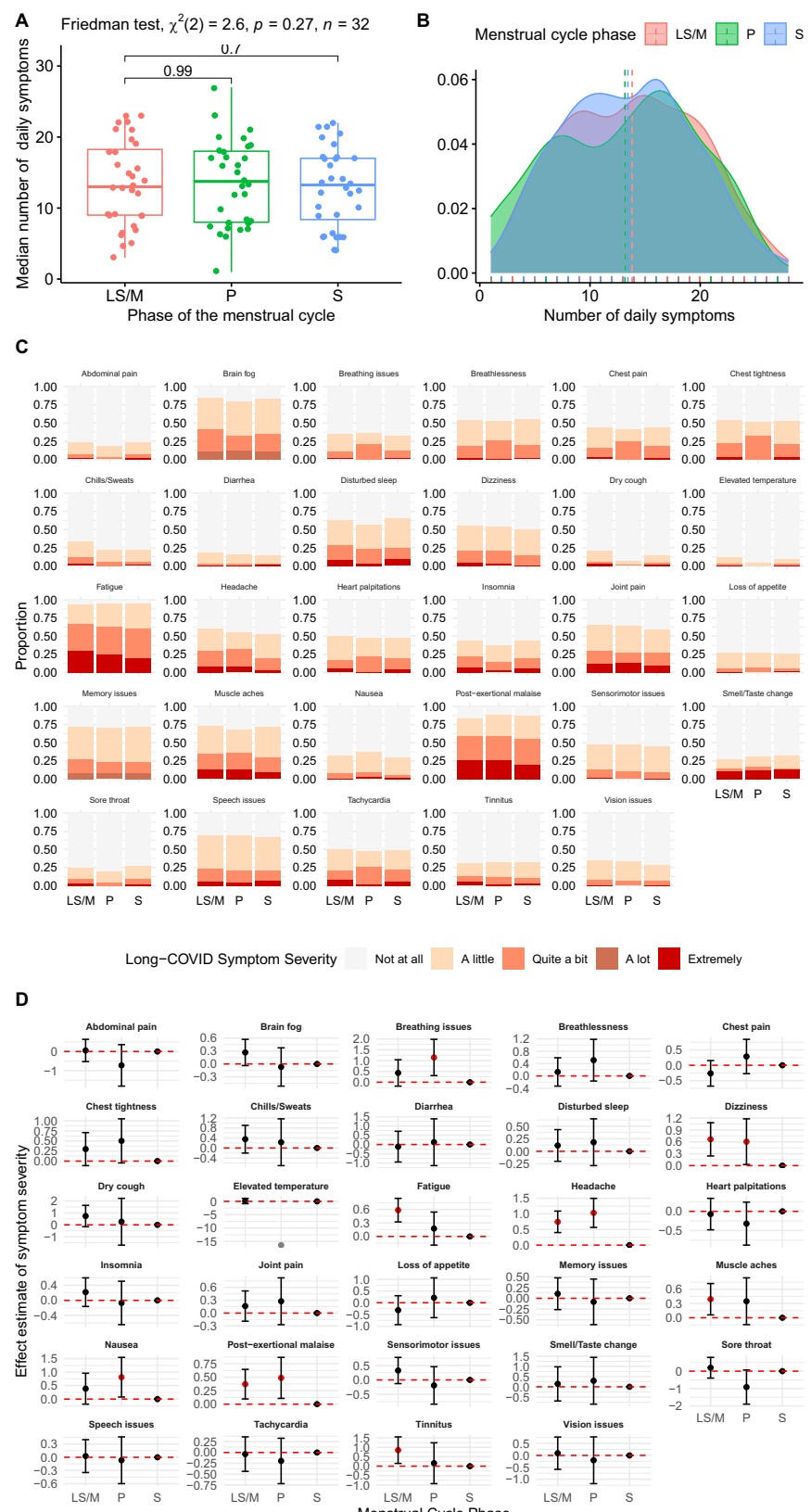

long COVID at three points in their menstrual cycle; proliferative, secretory and menstrual phases (Table 2, Fig. 2B).

Gold-standard steroid measurement by LC-MS/MS analysis was used to compare steroid serum levels from those with long COVID with samples collected from women prior to the COVID pandemic. Our group of women who had recovered fully from acute COVID were significantly younger than no COVID controls and those with long COVID and were therefore not included in ovarian hormone analyses (Table 2). A significantly higher level of serum 5α-dihydrotestosterone, the most active androgen, was observed in women with long COVID versus those with no COVID during the secretory phase ($p = 0.0138$, 95% CR = [−0.06732 to −0.007898], DF 75) (Fig. 4A). When comparing

**Fig. 3 | Prospective study of long COVID symptoms across the menstrual cycle.** **A** Boxplot representing the median number of daily symptoms across the three phases of the menstrual cycle. Box plots display the median (centre line), first quartile (Q1, lower box boundary) and third quartile (Q3, upper box boundary), with whiskers extending to the minimum and maximum values within 1.5×IQR of the box boundaries. Late Secretory/Menstrual (LS/M) phase: $n = 32$, minimum=3, Q1 = 9, median=13, Q3 = 18.25, maximum = 23; Proliferative (P) phase: $n = 32$, minimum=1, Q1 = 8, median=13.75, Q3 = 18.25, maximum = 23; Secretory (S) phase: $n = 32$, minimum=4, Q1 = 8.38, median=13.25, Q3 = 17, maximum = 22. The results of a two-sided Friedman test indicate no significant differences in the median number of symptoms between phases within individuals. **B** Density plot illustrating the probability distribution of symptom counts across the menstrual cycle phases. There was a large overlap in the distribution of the number of symptoms across the phases. **C** Distribution of symptom severity for each symptom across the phases of the menstrual cycle. **D** Predicted values and 95% confidence intervals for the association between menstrual cycle phase and the severity of long COVID-19 symptoms. Sample size: $n = 54$ participants (biological replicates) reporting across 29 symptoms, contributing 18,506 observations (technical replicates) over 930 tracking days. Menstrual phase distribution: S phase ($n = 44$ participants, 8143

observations); LS/M phase ($n = 54$ participants, 7751 observations); P phase ($n = 33$ participants, 2612 observations). Unit of study: individual participants with daily symptom tracking. Estimates derived from unadjusted univariable cumulative-link mixed models (one model per symptom) examining symptom severity across menstrual cycle phases. Cumulative odds ratios represent the odds of experiencing higher versus lower severity categories, with the secretory (S) phase as the reference category. Several symptoms showed significantly increased severity during specific menstrual phases: Late secretory/menstrual (LS/M) phase was associated with more severe dizziness (OR = 1.94, FDR-adjusted $p = 0.005$), fatigue (OR = 1.80, FDR-adjusted $p = 0.0000$), headache (OR = 2.11, FDR-adjusted $p = 0.0000$), muscle aches (OR = 1.48, FDR-adjusted $p = 0.038$), post-exertional malaise (OR = 1.45, FDR-adjusted $p = 0.016$), and tinnitus (OR = 1.42, FDR-adjusted $p = 0.034$) compared to the S phase. Proliferative (P) phase was associated with more severe breathing issues (OR = 3.14, FDR-adjusted $p = 0.015$), headache (OR = 2.81, FDR-adjusted $p = 0.0000$), post-exertional malaise (OR = 1.63, FDR-adjusted $p = 0.024$), with a trend toward more severe nausea (OR = 2.25, FDR-adjusted $p = 0.055$) and dizziness (OR = 1.83, FDR-adjusted $p = 0.072$) compared to the S phase. Red dots indicate statistically significant associations (Pr(>|z|) <0.05). Data and code are available on GitHub https://github.com/ataquette/Long-COVID-Mens.

**Table 2 | Participant characteristics for serum/endometrial tissue study (*$P = 0.0137$)**

| Participant characteristics mean (range) or number (%) | Control ($n = 40$) | COVID recovered ($n = 10$) | Long COVID ($n = 10$) | Long COVID endometrial subset ($n = 7/10$) |
|---|---|---|---|---|
| Age (years) | 40 (28–48) | 33 (24–45)* | 39 (29–45) | 40 (33–45) |
| Parity | 1 (0–4) | 1 (0–3) | 1 (0–2) | 1 (0–2) |
| Body mass index (kg/m2) | 28 (20–46) | 24 (19–31) | 30 (18–37) | 32 (28–37) |
| Cycle length (days) | 28 (22–33) | 27 (24–28) | 28 (25–31) | 28 (26–31) |
| Smoking | 13% | 0% | 0% | 0% |

those with long COVID and controls at each cycle stage there were no significant differences in 17β-estradiol or progesterone between the two groups. To examine ovarian hormone fluctuations across the menstrual cycle in those with and without long COVID, we analysed a subset of our control cohort who had provided paired samples from the secretory and menstrual phases ($n = 14$) and compared these with paired samples from the long COVID group ($n = 7$). This did not reveal any clear differences in how ovarian hormone levels changed from secretory to menstrual phase in those with long COVID versus controls (Suppl. Fig. 2A). We then examined ovarian sex hormones in endometrial tissue from the subset of women who provided endometrial biopsies, as levels in the local endometrial environment are often different to circulating hormone levels due to peripheral tissue metabolism. There were no significant differences in endometrial levels of estradiol, estrone, estrone/17β-estradiol (E1/E2) ratio, progesterone, testosterone or 5α-dihydrotestosterone between those with long COVID and controls at any cycle stage (Fig. 4B). Comparison of paired endometrial samples from the same woman across the menstrual cycle was not possible due to the smaller endometrial sample size. An unpaired two-way ANOVA revealed a significant changes in endometrial progesterone across the menstrual cycle in both long COVID and control groups, as expected. For endometrial testosterone, there were significant differences between menstrual and proliferative phases ($p = 0.0084$) in the control group, which were not demonstrated in the long COVID group.

Ovarian sex hormone receptors in endometrial tissue across the menstrual cycle were then examined by PCR (Fig. 4C). Progesterone receptor mRNA (*PGR*) was lower in those with long COVID versus controls during the proliferative phase of the menstrual cycle ($p = 0.0096$ 95% CI = [0.1671; 1.136], DF 42). Endometrial PGR immunohistochemical nuclear staining quantification revealed significantly lower % positive cells in those with Long COVID vs controls during the secretory phase but histoscores, combining number of cells stained

and the intensity of staining, were not different in those with long COVID and controls at any cycle stage (Suppl. Fig. 3). Given differences in circulating 5α-dihydrotestosterone levels, we performed immunohistochemistry for the androgen receptor (AR) and found localisation to the nuclei of stromal and occasional glandular epithelial cells (Fig. 4D). There was a high percentage of AR-positive endometrial cells during the proliferative phase (Fig. 4E), consistent with published literature[35,36]. Comparison of endometrial AR staining in those with and without long COVID revealed a significantly lower percentage of positive cells in those with long COVID during the menstrual phase (Fig. 4E) and a significantly lower histoscore during the menstrual and proliferative phases (Fig. 4F). We then examined if ovarian reserve was affected by long COVID and revealed serum Anti-Müllerian Hormone (AMH) was comparable between long COVID and controls (Fig. 4G).

**Inflammatory mediators are aberrant in serum and endometrium from women with long COVID versus controls across the menstrual cycle**

Serum cortisol levels can impact the inflammatory response and have previously been found to be significantly lower in individuals with long COVID versus controls[37]. In addition, local endometrial glucocorticoid deficiency has been associated with increased menstrual blood loss[38]. Hence, we first examined serum glucocorticoid levels in those who had never had COVID, women who had recovered from acute COVID and those with long COVID across the menstrual cycle. We found no differences in serum cortisol, cortisone or cortisol/cortisone ratio when comparing those with long COVID and these two control groups at any phase of the menstrual cycle, including in our sub-analysis of paired samples (Fig. 5A, Suppl. Fig. 2B). We then examined glucocorticoid levels in endometrial tissue. It was not possible to obtain endometrial biopsies from our 'healthy' COVID recovered group. There were no differences in endometrial glucocorticoids between those with long COVID and controls at any cycle stage. However, the cortisol/cortisone

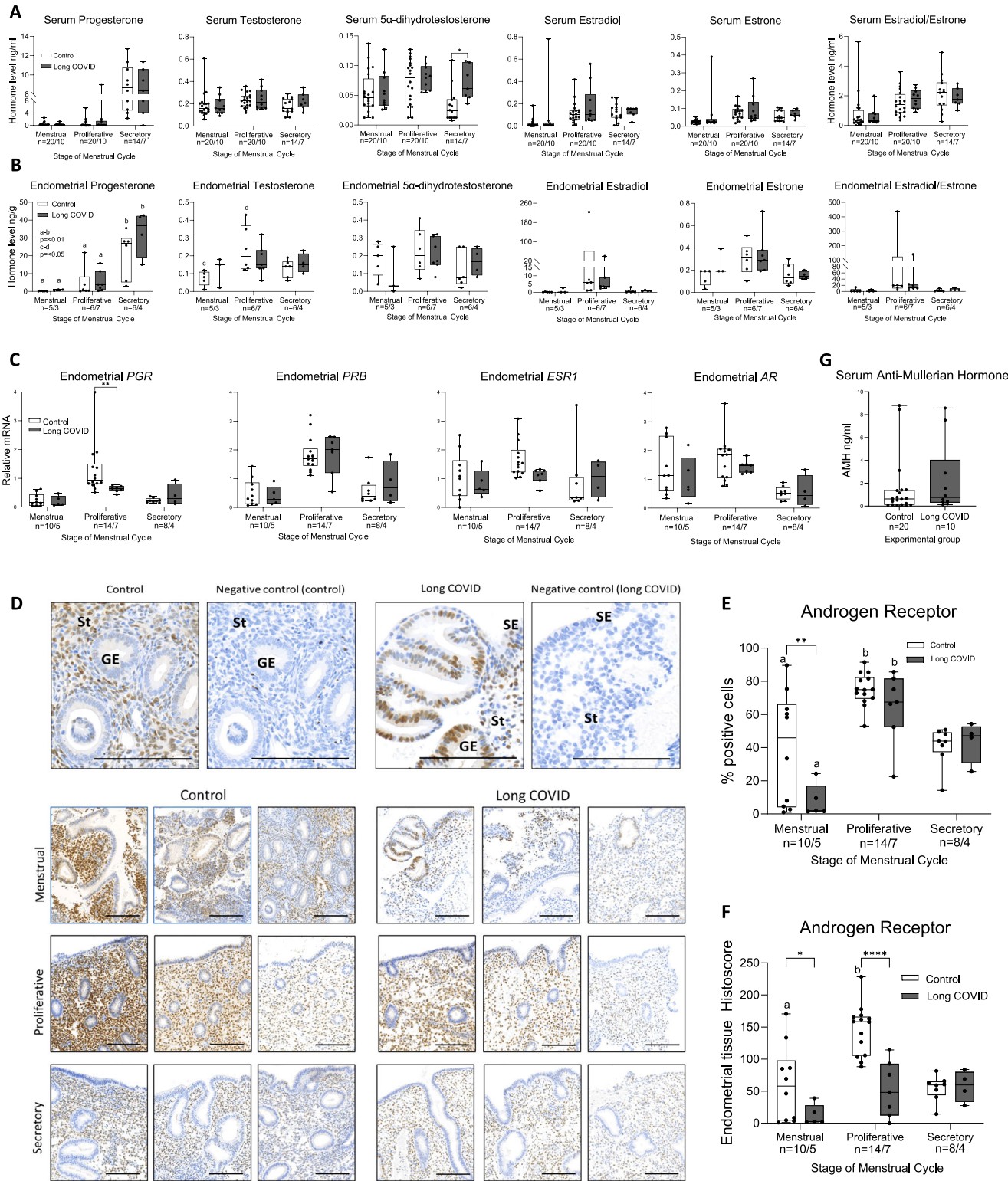

ratio was significantly higher at menstruation versus other cycle stages in controls (menstrual v proliferative $p = 0.0036$, menstrual v secretory $p = 0.0033$) but did not reach statistical significance in those with long COVID (Fig. 5B).

Long COVID has been associated with persistent systemic inflammation, including elevated markers of myeloid inflammation, complement activation and cytokines[39,40], but changes across the menstrual cycle have not been examined to date. We examined a panel of cytokines in serum at three stages of the menstrual cycle, comparing those who had never had COVID, those recovered from COVID and those with long COVID (Fig. 5C). Overall, we observed higher levels of serum cytokines in those with long COVID during the menstrual phase, and lower levels during the proliferative phase, when compared to those with no COVID or those who had recovered from acute COVID. Serum TNF was significantly higher during the menstrual phase in those with long COVID versus no COVID controls ($p = 0.0002$, 95% CI = [−5.650 to −1.509], DF 102) and compared to women who had recovered from COVID ($p = 0.0177$, 95% CI = [0.4016−5.182], DF 102).

**Fig. 4 | Ovarian sex hormones across the menstrual cycle in those with and without long COVID. A** LC-MS/MS revealed serum 5α-dihydrotestosterone was significantly higher in those with long COVID compared to controls in the secretory phase of the cycle ($p = 0.0138$). **B** There were no differences in local endometrial tissue ovarian sex hormone concentrations measured by LC-MS/MS in women with long COVID compared to controls. **C** There were significantly lower *PGR* mRNA concentrations in proliferative endometrial tissue from women with long COVID versus controls ($p = 0.0096$). **D** Immunohistochemistry staining for AR ($n = 3$ representative slides shown per group at each menstrual cycle stage) revealed less intense stromal staining during the menstrual and proliferative phases and more menstrual phase glandular epithelial staining in endometrium from those with long COVID. St stroma, GE glandular epithelium, scale bar = 100 μM. **E** Quantification of the percentage of endometrial cells positive for AR revealed fewer positive cells in menstrual endometrium from those with long COVID vs controls ($p = 0.0049$). **F** Endometrium from those with long COVID had a significantly lower AR histoscore (number and intensity of staining) during the menstrual ($p = 0.0335$) and proliferative ($p = <0.0001$) phases. **G** AMH levels in those with long COVID versus controls were similar. Box and whisker plots: the box represents the upper and lower quartiles, with the horizontal line representing the median, and the whiskers represent the minimum and maximum values. Dark grey bars: long COVID, white bars: samples collected from women prior to the COVID pandemic. Statistical test: two-way ANOVA with Tukey's multiple comparisons test, *$p < 0.05$, **$p < 0.01$, ***$p < 0.001$, ****$p < 0.0001$. Letters = significance across menstrual cycle, a, b $p < 0.01$, c, d $p < 0.05$. Source data are provided as a Source Data file.

Those with long COVID had lower serum TNF during the proliferative phase of the cycle vs those who had never had COVID ($p = 0.0197$, 95% CI = [0.3139; 4.454], DF 102). Serum IL8 protein was also significantly lower in those with long COVID versus no COVID controls during the proliferative phase of the menstrual cycle ($p = 0.0016$, 95% CI = [5.127; 25.81]). There were no significant differences detected in serum IFNG, IL10, or IL6 between the groups at any menstrual cycle stage. A similar non-significant trend was observed for serum IL6 protein, with higher menstrual and lower proliferative levels in those with long COVID. Insufficient levels of IL-2 were detected to allow comparison between the long COVID and control groups. Our subset analysis of paired serum samples from the secretory and menstrual phases showed significant increases in IL8 and TNF on transit into active menstruation in both no COVID and long COVID groups, but IL10 and IL6 serum levels were significantly increased during menstruation in the group with long COVID but not those who never had COVID or those who had recovered from acute COVID (Suppl. Fig. 2C).

Excessive local endometrial tissue inflammation has previously been observed in those with heavy menstrual bleeding[34,41], hence we also examined endometrial tissue cytokine mRNA concentrations to compare those with long COVID and no COVID controls at three points in the menstrual cycle (Fig. 5D). Highest endometrial expression of all cytokines examined was during menstruation, consistent with the local inflammatory process of menstruation[42]. Significantly lower menstrual endometrial *IL10* and *TNF* was detected in women with long COVID versus controls ($p = 0.0121$, 95% CI = [1.033; 7.929], DF 42 and $p = 0.0406$, 95% CI = [0.8572; 37.38], DF 42 respectively) and a non-significant trend towards lower menstrual *IL8* and lower *IL6* was also observed ($p = 0.0508$, $p = 0.1502$). Given the importance of endometrial immune cells at menstruation[43,44], we then localised endometrial neutrophils and macrophages in menstrual endometrium by immunohistochemistry. Staining of neutrophil marker LL37 revealed nuclear staining in occasional cells within the stromal compartment, particularly localised to areas undergoing breakdown (Fig. 5E). Quantification of immune cell staining revealed that menstrual endometrial neutrophil number was not significantly different in those with and without long COVID (Fig. 5F), but neutrophil aggregates were visible within endometrial glands in those with long COVID and were not visible in endometrium from those who had never had COVID (Fig. 5E). Endometrial macrophages, detected by immunohistochemical staining for CD68, were localised to the stromal cell compartment (Fig. 5G). Again, menstrual endometrial macrophage number was not significantly different in endometrium from those with and without long COVID, but there was evidence of accumulation within endometrial glands in endometrium from those with long COVID.

## Discussion
The bidirectional impact of COVID-19 on menstruation and of the menstrual cycle on symptoms of COVID-19 has been the subject of intense public interest, but research in this area has been lacking. To address this, we performed three complementary studies. Firstly, we revealed that menstrual duration, volume and intermenstrual bleeding are reported to be significantly increased in those with long COVID when compared to those who have never had COVID, whereas menstrual regularity and frequency were not different. Second, we showed that long COVID symptom severity was increased during the peri-menstrual phase in those with regular menstrual cycles. Finally, investigation of potential mechanisms underpinning these associations in a small group of well-characterised patients revealed that serum 5α-dihydrotestosterone was higher in the secretory phase and endometrial AR staining appeared less intense in the menstrual and proliferative phases in those with long COVID versus controls. Serum menstrual TNF was found to be increased in those with long COVID, with blood serum cytokine expression tending to be increased in those with long COVID versus controls during the menstrual phase and decreased in the proliferative phase. Collectively, these findings indicate a bidirectional relationship between long COVID and abnormal uterine bleeding, potentially mediated by disruptions in androgen regulation and the inflammatory response within the endometrium.

Abnormal uterine bleeding (AUB) is extremely common, affecting up to one in three women of reproductive age across the globe[3,4,45] and causing a significant negative impact on physical, mental and financial wellbeing. Heavy menstrual bleeding is a major contributor to anemia in women of reproductive age, but is often overlooked[46,47]. Our finding that long COVID is significantly associated with AUB and may increase the frequency of these already highly prevalent symptoms highlights the potential additional burden on women, those who menstruate and healthcare services. We acknowledge that survey recruitment and retrospective reporting of symptoms may introduce sources of bias. However, the title of our survey was purposefully broad, 'Women's reproductive health and the COVID pandemic', to limit oversampling of participants with a specific interest in menstrual cycles. In addition, our survey was conducted before widespread media attention to the topic of menstrual disturbances (April in the US and May 2021 for the BBC), although we acknowledge that reporting of menstrual changes in those with long COVID may have occurred earlier. Furthermore, we fine-tuned advert targeting (to the extent that Facebook allows) throughout the campaign and used a stratified sampling strategy to ensure that subgroups of the UK population in terms of age, income and ethnicity were represented in the final sample. We found that the initial survey was broadly representative of people infected with COVID in the UK (8.9% with a positive PCR test in our study compared to a national proportion of 6.6% at the time). Despite these measures, survey participants were predominantly white, educated women. This reflects those with the capacity to participate in such studies, with all reproductive-aged women being an underrepresented group in research due to caring responsibilities and work. Women from ethnic minorities and lower socioeconomic groups will face additional challenges. Strategies to mitigate such challenges were limited during the COVID pandemic, and we therefore advise caution when extrapolating findings across different populations.

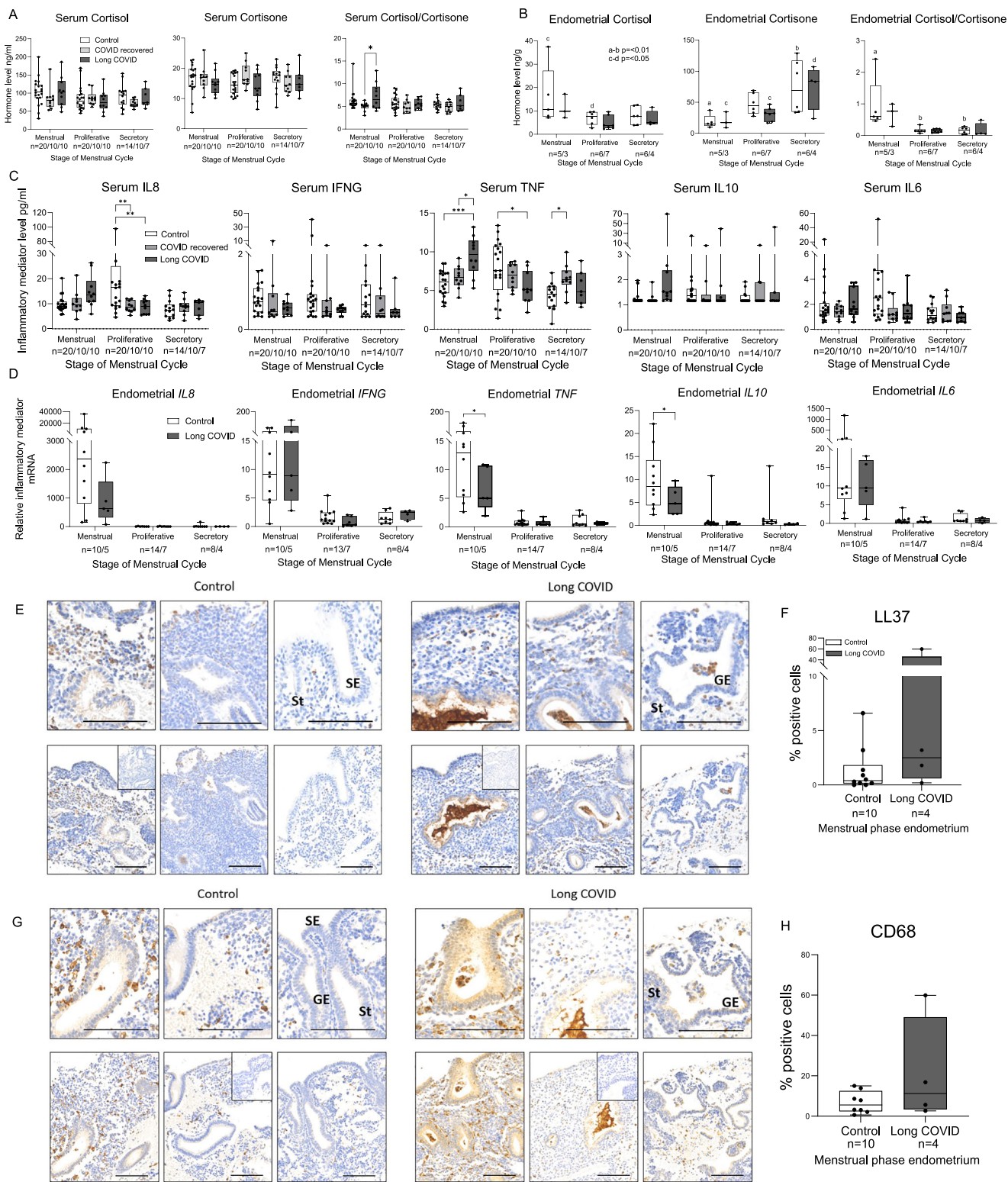

During analysis, we carefully considered and adjusted for potential confounders, such as pre-existing abnormal menstrual parameters, exogenous hormone use and formal diagnosis of conditions that affect reproductive function. Similar proportions of those with no COVID, acute COVID, and long COVID had been vaccinated (Table 1). Our previous analysis of those who were vaccinated also revealed that, although menstrual disturbance was reported in almost 20% after vaccination, the risk of developing AUB was not significantly different from those who were not vaccinated[22]. With long COVID estimated to

affect 3–7% of the global population[48,49], the increased rates of menstrual volume and duration reported by those who menstruate will predispose them to anemia and potentially exacerbate troublesome long COVID symptoms. Indeed, anemia and low serum iron were associated with long COVID, and maldistribution of iron was observed in those with persistent symptoms[50]. Authors proposed that infection-triggered defects in iron homoeostasis increased the likelihood of developing long COVID, but did not examine menstrual parameters in their female participants. Herein, we reveal that acute and long COVID

**Fig. 5 | Inflammatory profiles are altered in women with long COVID versus controls. A** Serum cortisol and cortisone measured by LC-MS/MS were not different between those who had never had COVID (Control), those recovered from acute COVID (COVID recovered) and those with long COVID at any point of the menstrual cycle. Serum cortisol/cortisone ratio was significantly higher in the menstrual phase in those with long COVID versus the COVID recovered group ($p = 0.0113$). **B** There were no differences in cortisol and cortisone levels in endometrial tissue from women with long COVID versus controls at any cycle stage. Cortisol/cortisone ratio in the local endometrial tissue from controls displayed a significant increase in this ratio at menstruation (menstrual versus proliferative phase $p = 0.0036$, menstrual versus secretory phase $p = 0.0033$) that did not reach statistical significance in those with long COVID. **C** Serum TNF levels were higher in samples from women with long COVID during the menstrual phase (long COVID versus pre-pandemic control $p = 0.0002$, long COVID versus COVID recovered $p = 0.0177$). Serum TNF and IL8 were lower in the proliferative phase in those with long COVID when compared to controls (TNF $p = 0.0197$; IL8 $p = 0.0016$). **D** Endometrial tissue *TNF* and *IL10* mRNA concentrations were significantly lower in

women with long COVID during menstruation (*TNF p* = 0.0406, *IL10 p* = 0.0121). **E** Immunohistochemical staining of neutrophils with LL37 in menstrual endometrial tissue ($n = 3$ controls and $n = 3$ long COVID) revealed the presence of neutrophil aggregates within the glandular epithelium of endometrium from those with long COVID (top row: high power, bottom row: lower power, inset: negative control). **F** Quantification of the percentage of LL37 positively stained cells revealed no significant differences in endometrium from those with long COVID versus no COVID. **G** Immunohistochemical staining of CD68 in menstrual endometrial tissue revealed glandular aggregates were present in those with long COVID. (top row: high magnification, bottom row: lower magnification, inset: negative control). **H** There were no significant differences in macrophage number between those with long COVID ($n = 3$) and controls ($n = 3$) on quantification. GE glandular epithelium, St stroma, scale bar = 100 mM. Statistical analyses were performed by two-way ANOVA with Tukey's multiple comparisons test, $*p < 0.05$, $**p < 0.01$, $***p < 0.001$. Box and whisker plots: the box represents the upper and lower quartiles, with the horizontal line representing the median, and the whiskers represent the minimum and maximum values. Source data are provided as a Source Data file.

resulted in increased reports of AUB, which could additionally predispose to iron deficiency and anemia. We were unable to measure iron or haemoglobin levels in survey or app study participants as these studies were completed remotely; therefore, this link remains theoretical. Nonetheless, this may provide one explanation for the increased incidence of long COVID in women, particularly those under 50 years old[29]. Hence, correction of iron deficiency and timely, effective treatment of AUB symptoms should be part of holistic care.

Examination of the number, presence and severity of long COVID symptoms across the menstrual cycle in reproductive-aged women revealed symptom severity was increased after progesterone withdrawal during the peri-menstrual and proliferative phases. We limited participant bias during this study by using clear questions based on patient-reported symptoms[28], but acknowledge that recruitment bias may be present. We also acknowledge that the number of participants is relatively small. Strengths of the study include real-time data collection and that each participant with long COVID served as their own control, with repeated measures across the menstrual cycle. This menstrual cycle effect may not be specific to symptoms of long COVID, and women without long COVID may also experience menstrual cycle variations with other symptoms. This does affect the clinical relevance of our findings, which are consistent with a hormonal influence on long COVID symptom severity.

To direct appropriate treatment of the symptoms of AUB, delineating the underpinning mechanisms is essential. Given the onset of menstrual disturbance in those with acute and long COVID, nonstructural AUB is most likely, rather than the development of structural pathologies such as fibroids or adenomyosis. This non-structural AUB may be of ovarian or endometrial origin. Co-expression of *ACE2* and *TMPRSS2* has been detected using scRNA-Seq in human ovarian tissue[51–53], indicating the potential for COVID-19 to affect ovarian function. In contrast, endometrial expression of *ACE2* and *TMPRESS2* was low[54,55]. Reassuringly, we did not observe significant alterations in serum estradiol or progesterone levels in those with long COVID, consistent with previous studies examining those with acute COVID[23]. Unfortunately, it was not possible to compare ovarian hormone levels in our COVID recovered group due to their significantly younger age, which markedly affects ovarian hormone levels. Our finding of no significant changes in ovarian hormone levels in those with long COVID versus age-matched controls who never had COVID is consistent with our survey data, where those with long COVID did not have statistically significant changes in menstrual frequency or regularity compared to those with no COVID. AMH levels were also comparable in our long COVID group and age-matched controls who had never had COVID. These results contrast with the reduced gonadal hormone levels observed in male COVID-19 patients and the GnRH neuronal death observed in post-mortem patient brains[56]. Several factors may account

for this discrepancy. Firstly, sex-based differences may be present, with the study by Sauve et al. excluding female patients "due to the difficulty of following the menstrual cycles of female patients". Second, the severity and timing of COVID-19 were different between studies, with this previous study examining those admitted to intensive care with acute COVID and post-mortem specimens. Acute illness is known to temporarily downregulate the Hypothalamic-Pituitary-Gonadal (HPG) axis, and this may not be apparent in those with long COVID. Finally, we were forced to limit our repeated, timed sample collection to those with long COVID and regular menstrual cycles for practical reasons, with our clinical recruitment occurring at a time of significant public and hospital restrictions in a cohort of patients with debilitating symptoms. This meant that those with significant suppression of the HPG axis were excluded, but allowed examination of carefully categorised repeat samples from across the menstrual cycle, although we acknowledge that our numbers are low. A further limitation of our study is that it was performed in predominantly white women, and our findings should be applied with caution to a non-white population.

Consistent with the normal 17β-estradiol and progesterone serum levels we observed in those with long COVID, serum testosterone levels were also similar between the groups. However, 5α-dihydrotestosterone, the most active androgen, had higher levels in those with long COVID during the secretory phase. Androgens impact the proliferation, migration and survival of stromal cells within the human endometrium[57]. In a mouse model of simulated menses, administration of a subcutaneous injection of 5α-dihydrotestosterone prior to menstrual onset led to delayed repair of the menstrual endometrium and prolonged simulated menstrual bleeding that deserved further investigation[58]. Therefore, our finding of increased serum 5α-dihydrotestosterone in the secretory phase of those with long COVID provides a potential mechanism that could alter endometrial function during menstruation and result in prolonged, heavy menstrual bleeding. 5α-reductase drives peripheral conversion of testosterone to 5α-dihydrotestosterone. Therefore, our findings are consistent with more active 5α-reductase in women with long COVID. It remains to be determined if long COVID causes this increase in activity or if those with more active 5α-reductase are more susceptible to developing long COVID, e.g., those with polycystic ovary syndrome[51,59]. All participants in our study had regular menstrual cycles, and none had a confirmed diagnosis of PCOS, but undiagnosed PCOS cannot be excluded. Regardless, this finding highlights a potential therapeutic pathway for further investigation, to examine the impact of 5α-reductase inhibitors on long COVID onset, symptoms and COVID-associated AUB, acknowledging the challenges of their use in reproductive-aged women.

We then examined a panel of cytokines in serum and endometrium from those with long COVID, who had recovered from COVID

and who had never had COVID. The endometrium is susceptible to alterations in systemic inflammation due to the key role of circulating immune cells in the breakdown and repair of the endometrium at menstruation, in concert with tissue resident immune cells[60]. We revealed an altered serum cytokine profile in those with long COVID that varied across the menstrual cycle, with higher cytokine levels during menses and lower levels during the proliferative phase. This was not observed in those who had recovered from acute COVID. Other studies have found differences in soluble immune mediators in those with long COVID[37,61], but none have examined changes across the menstrual cycle. Our data suggest that adjustment for phase of the menstrual cycle will be necessary in future studies of serum cytokine levels in reproductive-aged females. Furthermore, this cytokine dysregulation merits further investigation as a potential mechanism for the increased severity of long COVID symptoms reported immediately prior to and during menstruation. It remains to be determined if anti-inflammatory therapies administered pre-menstrually or the prevention of sex hormone withdrawal in the late secretory phase with exogenous hormones can prevent excessive inflammation and an increase in long COVID symptom severity.

Cortisol is also known to impact the immune response and has recently emerged as a potential biomarker for long COVID, with lower levels in long COVID patients versus controls[37]. Our smaller study here did not detect any such differences, consistent with findings by Fleischer et al[62]. In fact, those with long COVID had a higher cortisol/cortisone ratio during the menstrual phase. However, this may reflect increased adrenal hormone production with age, with participants who had recovered from COVID being younger than the long COVID/No COVID age-matched participants. Of note, our study was not designed to analyse cortisol levels, and these single samples were not consistent for time since waking, a limitation shared by the two previous studies[37,62]. With this caveat, we did not see any significant differences in serum cortisol levels across the menstrual cycle, indicating relative stability with ovarian hormone flux.

There is evidence that the endometrium displays classic features of inflammation at menstruation, including release of prostaglandins, increased permeability of blood vessels and an abundance of leukocytes[42]. In addition, there is evidence of excessive inflammation during menstruation in those with AUB[6]. Glucocorticoids have an anti-inflammatory effect that is mediated by cortisol, and women with heavy menstrual bleeding have previously been found to have higher levels of 11β-HSD2, which inactivates cortisol[63]. Oral dexamethasone, a synthetic glucocorticoid, was shown to have therapeutic benefit in the management of HMB when administered in the secretory phase[38]. Therefore, we examined endometrial levels of cortisol and cortisone in those with and without long COVID and found that the endometria from those with long COVID lacked the significant increase in cortisol/cortisone ratio that was observed in controls at menstruation. This highlights a potential therapeutic benefit of dexamethasone in those with long COVID who are experiencing AUB that merits further investigation.

Previous studies comparing endometrial tissue collected from those with and without heavy menstrual bleeding have revealed increased levels of TNF, *COX-2* and prostaglandins[34,41,64]. Indeed, non-steroidal anti-inflammatory medications taken at the time of menstruation are an effective treatment for heavy menstrual bleeding[65]. In contrast, we found that endometrial tissue from those with long COVID had lower levels of *TNF* when compared to controls. However, the immunosuppressive cytokine *IL10* was also lower during menstruation in those with long COVID. To examine the impact of this altered cytokine environment, we examined endometrial neutrophil and macrophages and found their number appeared similar between the groups, but neutrophil aggregates were visualised in endometrial epithelial glands from those with long COVID. It remains to be determined if these are neutrophil extracellular traps, which are well known

to have the ability to damage tissue[66]. Differences in neutrophil degranulation and high neutrophil activity have been identified in blood samples from those with long COVID versus those who had fully recovered[67]. In addition, endometrial neutrophils are important for endometrial function at menstruation[43,68]. Therefore, the presence of altered neutrophils in the endometrium at this time has the potential to contribute to the AUB reported by many women with long COVID.

In conclusion, this study provides evidence of an association between long COVID and AUB that may be the result of increased androgens and an altered endometrial inflammatory response at menstruation, warranting further investigation. Reassuringly, ovarian function appears to be maintained in this group of regularly cycling women with long COVID. We also reveal an association of increased severity of long COVID symptoms with the late secretory/menstrual phase of the menstrual cycle, when progesterone levels rapidly decline. This may be explained by increased cytokine production during the menstrual phase, which was greater in those with long COVID than in controls. Based on our findings, we recommend future investigation of specific treatments for AUB in those with long COVID, consideration of the menstrual cycle in future long COVID biomarker development, and a focus on female-specific treatments for long COVID.

## Methods

### Online COVID and reproductive health survey

The study, titled "The COVID-19 Pandemic and Women's Reproductive Health" received a favourable ethical opinion from the Oxford University School of Anthropology and Museum Ethnography Departmental Research Ethics Committee [SME_C1A_20_029]. The survey was designed to evaluate whether and how the COVID-19 pandemic influenced menstrual health. We incorporated feedback from women suffering from long COVID via long COVID Support (https://www.longcovid.org/). An online survey was launched on March 8, 2021, and was hosted on the Qualtrics platform (www.qualtrics.com). All survey responses were anonymized using randomly generated IDs. This allowed the collection of retrospective and self-reported data on menstrual cycles, behaviour, life circumstances and health before and during the pandemic, as well as COVID-19 disease and vaccination history. The survey included a maximum of 105 questions depending on individual circumstances and took an average of 24 minutes to complete. After consent, 61% of eligible participants answered all questions (on average, participants completed 80% of the questionnaire). To minimise survey fatigue, progress could be saved for up to 14 days to allow participants to resume later. The survey ran from 08/03/21 to 01/06/21 and was closed when there had been no new entries for a week.

Participant eligibility criteria included age >18, having ever menstruated, currently living in the UK, and giving informed consent to the use of their data. The survey was written in English and disseminated through a Facebook advertising campaign targeting all menstruators in the UK, and included images of women of diverse ethnicities, ages, and abilities, as well as images of breastfeeding and pregnant women. The title of the survey was kept general ("women's reproductive health and the COVID pandemic") so as not to over-sample individuals with a specific interest in menstrual cycles and COVID infection or vaccination. We fine-tuned the ad targeting (to the extent that Facebook allows) throughout the campaign to ensure an even geographical and socioeconomic spread. We also used a stratified sampling strategy to ensure that subgroups of the UK population in terms of age, income and ethnicity were represented in the final sample. In total, 695,543 people viewed the survey ad on their Facebook page, and 26,710 with the eligible criteria gave consent and completed it (there were no duplicates), leading to a 3.8% response rate. In this sample, participants were aged 18–45, 95% identified as being of White ethnicity and 99% identified as women.

## Menstrual parameters

We operationalized our outcome variables to approximate the FIGO classification system for normal and abnormal uterine bleeding in relation to five parameters: frequency, regularity, duration, volume, and inter-menstrual bleeding (FIGO System 1)[1].

**Frequency.** In the latter part of the survey, participants were asked, "Over the last year, how many days long, on average, was your cycle (between the start of one bleed and the start of the next bleed)?". Based on the number of days reported, we created a variable with 3 possible outcomes (Normal [24–38 days], Frequent [<24 days], Infrequent [>38 days], based on FIGO definitions).

Participants were also asked "Over the last year, have your periods stopped?" and "Over the last year, did you miss your periods at least once?" Although "stop" and "miss" were not defined, concerns over "missing periods" were being reported on social media and thus this variable was meant to capture people's perception of their cycles from which we created a binary variable (perception of 'missing' or 'stopped' periods (0/1)).

**Regularity.** Participants were asked, "Over the last year, how irregular were the length of your menstrual cycles on average?". We created a variable with 3 possible outcomes (Normal[<2 days; 2–5 days; 5–10 days], Somewhat irregular [10–20 days], Very irregular [>20 days]).

**Duration.** Participants were asked, "Over the last year, have you noticed any changes in the length of your menstrual cycle? Days of bleeding (Period length)" We created a binary variable with two possible outcomes (Normal ≤8 days; Prolonged >8 + days]).

**Volume.** "Over the last year, have you noticed any changes in your periods?" There were four possible outcomes ("Heavier", "Lighter", "No Changes" and "Heavier and Lighter").

**Intermenstrual bleeding.** Over the last year, have you noticed any changes in spotting mid-cycle? There were four possible outcomes ("No changes", "More", "Sometimes", "Sometimes less and sometimes more".

## Exposures

The dataset included socio-demographic variables (age, income, education, gender, ethnic group, marital status, parity), standard proxies for health (BMI, smoking status, physical activity, regular use of vitamins/supplements, regular use of medicine) and reproductive variables indicative of menstrual health before the pandemic (age at menarche, cycle length, period length, cycle irregularity, heavy bleeding, exogenous hormone use and a formal diagnosis of conditions known to affect menstruation, e.g., Endometriosis, Polycystic ovary syndrome, HIV/AIDS, underactive thyroid, overactive thyroid, uterine polyps, uterine fibroids, eating disorders, interstitial cystitis, other), as well as vaccine-related, COVID and pandemic-related variables. COVID-19 disease was operationalized based on whether people thought they had had COVID, as widespread testing had not been available in the UK in the early months of the pandemic that fell within the survey period, leading to three categories: No COVID (no tests or negative tests), acute COVID (symptoms lasting less than 28 days) and long Covid (symptoms lasting more than 28 days; we only included people who had symptoms at least a month before taking up the survey). Exogenous hormone use was categorised as progestogen-only (hormonal coil or IUS, implant, injectable, progestogen-only pill), combined estrogen and progestin (the pill, the patch, vaginal ring), copper IUD, sterilization, none (fertility awareness, condom, female condom, diaphragm) and other (e.g., oral non-contraceptive progestins).

## Statistical analysis

R version 4.4.2 (2024-10-31) was used for statistical analysis. We restricted all analyses to pre-menopausal individuals living in the UK who had a period in the 12 months preceding the survey and who were not pregnant or breastfeeding. Further, we only included individuals who knew their COVID-19 disease and vaccination history at the time of the survey. We reported prevalence ratios and relative risk ratios in the text and plotted predicted probabilities from adjusted models to represent absolute effects adjusted for confounders. Our main exposure variable described participants' self-reported COVID-19 disease history and had 3 levels[1] No COVID[2]; Acute COVID and[3] Long COVID. Our referent group was "No COVID". We used multinomial models when the outcome variables were nominal (two or more categories with no intrinsic order) and log-binomial regressions when the outcome was dichotomous. To evaluate changes between menstrual cycle characteristics, we adjusted all models for menstrual symptom parameters before the pandemic, and included age, BMI, hormonal contraceptive use and presence of diagnosed reproductive disease at baseline as confounders as per hypothesised directed acyclic graphs (https://github.com/ataquette/Long-COVID-Mens). Estimates and confidence intervals on the log-odds scale were converted to relative risk or risk ratios (multinomial models), and those on the log-probability scale (log-binomial models) were converted to prevalence ratios for reporting in tables and figures. To investigate if any associations between our exposure variable and menstrual cycle changes were influenced by confounders, we compared models with and without interaction effects using the Akaike Information Criterion (AIC). We reported variables significant at the false discovery rate (FDR) threshold (FDR-corrected $p < 0.05$)[69].

## Missing data

The analysis of complete cases only by dropping missing cases can introduce bias and lead to a substantial reduction of statistical power[70], especially if it is plausible that the data are not missing at random or not completely at random. To handle missing data, we used a multiple imputation approach using the R package 'missRanger', which combines random forest imputation with predictive mean matching. Prior to all analyses, we imputed 5 datasets, with a maximum of 10 iterations specified for each imputation. Each imputation was also weighted by the degree of missing data for each participant, such that the contribution of data from participants with higher proportions of missingness was weighted down in the imputation. We set the maximum number of trees for the random forest to 200, but left all other random forest hyperparameters at their default. The average out-of-bag error rate for multiple imputation across all imputed datasets was 0.08 (range: 0–0.77). Parameter estimates for all five datasets were pooled to provide more accurate estimates.

## Prospective long COVID symptoms study across the menstrual cycle

Participants were recruited on social media via UK long COVID support groups, X and Instagram. All participants provided informed consent prior to taking part in the study and for their anonymized data to be transferred from the Balance App to the French National Centre for Scientific Research (CNRS) for analysis. The study was reviewed by CNRS and received GDPR regulation approval (DPD 2022/10). Recruitment occurred from 01/02/2022 until 13/04/2022, after which no new entries were recorded for at least 2 weeks.

First, participants completed an electronic consent followed by baseline questionnaire at recruitment to record data on socio-demographics (age, gender, residence, height and weight, education, working hours, number of deliveries, caring responsibilities, hormonal contraceptive use, ethnic group, smoking, medication, date of last period, vaccination and covid disease histories, presence and severity of symptoms of long COVID).

Second, participants received daily email links through a reproductive health app (https://www.balance-menopause.com/) to complete a daily survey to record their COVID symptoms and severity over 2 months. The daily questionnaire asked about menstrual volume, with the options of 'no bleeding', 'spotting', 'light bleeding', 'moderate bleeding' or 'heavy bleeding' before asking participants to rate how they felt each day on a scale of 0–100 (100 being how they felt pre COVID). They were then asked to rate how 29 common long COVID symptoms were affecting them on that day, 0: not at all, 1: a little, 2: quite a bit, 3: a lot, 4: extremely. These symptoms were chosen because previous research suggested that they were experienced by at least 40% of those with long COVID[28] and included brain fog/cognitive dysfunction, memory impairment, speech and language symptoms (e.g., difficulty finding words), sensorimotor symptoms (e.g., tingling/pins and needles), dizziness and balance issues, change in smell and taste, insomnia (unable to sleep), headache, disturbed sleep, fatigue, post-exertional malaise (exhaustion after exercise/effort), chills/flushing/sweats, elevated temperature (between 37 C and 38 C or 98.6 F and 100.4 F), heart palpitations (extra awareness of heart beat or irregular heart beat), tachycardia (heart beating too fast), pain/burning in the chest, chest tightness, muscle aches, joint pain, sore throat, vision symptoms (e.g., blurred vision, flashing), tinnitus (ringing in the ears), shortness of breath, dry cough, breathing difficulty, diarrhoea, loss of appetite, nausea, abdominal pain (excluding period pains).

We considered the start of a cycle to be a day marked by spotting or bleeding, where the previous day had no bleeding or spotting, and the following day also showed bleeding or spotting. We examined three outcomes (i) the number of daily COVID-19 symptoms experienced, which corresponds to the number of symptoms experienced each day with a severity score >0 (ii); the presence of each COVID-19 symptom and (iii) the severity of each COVID-19 symptom. Our main exposure was the phase of the menstrual cycle, classified into 3 levels: (i) the *perimenstrual* phase, which includes the 2 days before menstruation and all subsequent days of menstruation; (ii) the *proliferative* phase, from the first day without bleeding until 15 days before the next cycle; and (iii) the *secretory* phase, from 14 days to 3 days before the next cycle.

To analyse the number of daily symptoms, we first ran non-parametric Friedman tests for paired data to compare the median number of daily symptoms across the three distinct phases of the menstrual cycle, in a sample of individuals who contributed data to all three phases ($n = 32$). We then included all data and ran a minimally age-adjusted mixed Poisson regression routine using the package lmer4 to account for an unbalanced dataset, as well as repeated daily measures of symptoms across individuals and phases. Age was transformed into a category variable following the median, (<below 40 years, 40 years+). The model was checked for normality of residuals and overdispersion.

To analyse the presence of each symptom on a given day, we ran a series of multilevel binomial regressions adjusted for age. To analyse the severity of each symptom experienced, we conducted a cumulative link mixed model for ordinal outcomes. It was not possible to adjust for age or other covariates due to the limited number of points for some symptoms. Estimates and confidence intervals on the log scale were converted to rate ratios (Poisson model) and those on the log-probability scale (log-binomial models) were converted to risk ratios for reporting in tables and figures. We reported variables significant at the FDR threshold (FDR-corrected $p < 0.05$)[69].

## Biological sample collection from those with long COVID and controls

Blood serum and an optional endometrial biopsy were collected from 10 participants with longer term symptoms of COVID (present for more than 4 weeks, based on consensus definitions at the time of recruitment) after informed consent and with a favourable ethical opinion from East Midlands-Leicester South Research Ethics

Committee (21/EM/0166), Fig. 1. Participants responded to a study poster on social media and disseminated via Scottish online support groups for those with long COVID. Participants were 33–45 years old, had regular menstrual cycles (24–38 days) and reported ethnicity as White (English/Scottish/Welsh/Northern Irish, $n = 8$) or mixed/multiple ethnic group ($n = 2$) (Table 2). Four participants had previously had positive PCR tests for SARS-CoV, and the remaining six had symptoms confirmed to be consistent with acute infection by a healthcare professional. All participants received COVID vaccinations (from various manufacturers, $n = 7$ had received three doses, $n = 2$ had received 2 doses and $n = 1$ had received 1 dose). No participants were using exogenous hormones or an intrauterine device during the study or in the preceding 2 months. No participants were taking anti-coagulant medications. Those with known fibroids >3 cm, reproductive tract cancer, currently breastfeeding or using oral steroids were excluded. On average, participants had experienced long COVID symptoms for ~15 months at the time of study consent. Participants were asked which symptom was most troubling, and this was reported as fatigue in 5/10 participants.

Participants attended NHS Lothian, Scotland, UK, between November 2021 and April 2023 on three occasions; during menstruation, within a week of menstruation cessation and 7 days prior to the typical cycle length, aiming for the mid-secretory phase. Serum samples were collected and confirmed as menstrual ($n = 10$), proliferative ($n = 10$), and secretory ($n = 7$) (Fig. 2B), based on cycle length and date of last menstrual period. Two samples timed to capture the secretory phase were excluded due to likely anovulation on analysis of serum estradiol and progesterone on the day of the biopsy. The rates of anovulation in this study were similar to our previous studies prior to the COVID pandemic. Time of sample collection was between 0900 and 1200 for 17 samples (63%), between 1200 and 1700 for nine (33%), and not recorded for one (4%).

Endometrial biopsies were collected using an endometrial sampler in the outpatient department, with stage of cycle confirmed by consistency across three parameters of (i) last menstrual period, (ii) serum estradiol and progesterone at time of biopsy and (iii) histological grading by a consultant pathologist[71]. Samples were confirmed as menstrual ($n = 5$), proliferative ($n = 7$) and secretory ($n = 4$) (Fig. 1). Regarding timing of endometrial samples, seven were collected between 0900 and 1200 (44%) and nine between 1200 and 1700 (56%). Immediately after collection, tissue was divided and placed in RNA-*later*™ stabilization solution [Invitrogen by ThermoFisher Scientific, Leicestershire, UK, AM7020] and stored at −80 °C for RNA extraction and neutral buffered formalin prior to paraffin wax embedding.

Control blood serum samples ($n = 54$) and endometrial samples ($n = 32$) were selected from the Female Reproductive Tract Tissue Resource (20/ES/0119), limited to those collected from women ($n = 40$) prior to November 2019 (pre-COVID-19 pandemic) and matched for age and parity. The time of serum collection from control participants was between 0900 and 1200 in 26 (48%) and between 1200 and 1700 in 19 (35%) samples, with no time recorded for nine samples (17%). Endometrial samples were collected between 0900 and 1200 for $n = 16$ (50%) and between 1200 and 1700 for $n = 15$ (47%), with no time recorded for $n = 1$ (3%) endometrial sample.

An additional group ($n = 10$) of women who had fully recovered from previous SARS-CoV-2 infection (confirmed by lateral flow or PCR test) was recruited. Serum blood samples were provided on three occasions across the menstrual cycle after informed consent and a favourable ethical opinion (24-EMREC-015). Participants responded to a study poster disseminated via university mailing lists and were 24–45 years old, had regular menstrual cycles and reported ethnicity as white ($n = 9$) or mixed ($n = 1$) (Table 2). All participants received COVID vaccinations from various manufacturers, $n = 1$ had received 5 doses, $n = 1$ had received 4 doses, $n = 4$ had received 3 doses, $n = 2$ had received 2 doses, and $n = 1$ had received 1 dose. Similar to long COVID

**Table 3 | Summary of serum hormone levels for control and long COVID groups at three stages of the menstrual cycle**

| Stage of cycle | Estradiol (pmol/L), mean (range) | | | Progesterone (nmol/L), mean (range) | | |
|---|---|---|---|---|---|---|
| | Control | COVID recovered | Long COVID | Control | COVID recovered | Long COVID |
| Menstrual | 145.4 (20.0–548.0) | 180.3 (89.5–279.9) | 346.0 (20.0–2789.0) | 1.1 (0.2–3.4) | 3.3 (0.6–15.0) | 0.8 (0.2–3.9) |
| Proliferative | 403.8 (88.0–1331.0) | 599.1 (314.0–1064.0) | 578.7 (20.0–1706.0) | 4.1 (0.2–24.6) | 2.3 (0.7–11.8) | 1.9 (0.2–6.8) |
| Secretory | 520.8 (249.0–1226.0) | 507.3 (145.6–1155.0) | 411.5 (20.0–676.7) | 37.9 (13.4–68.2) | 46.4 (3.5–85.7) | 33.5 (13.2–44.5) |

recruitment, no participants were using exogenous hormones during the study or in the preceding 2 months, and those with reproductive tract cancer, currently pregnant or breastfeeding or using oral steroids were excluded. Of note, it was not possible to match this acute COVID group to the long COVID group for age, BMI or parity. Serum samples were confirmed as menstrual ($n = 10$), proliferative ($n = 10$), and secretory ($n = 10$) (Fig. 2B), based on cycle length, date of last menstrual period and estradiol and progesterone levels at time of sample provision. The time of serum collection was between 0900 and 1200 in 15 (50%) samples and between 1200 and 1700 in 15 (50%) samples.

**Immunoassay**
Blood serum samples were measured for ovarian sex hormone (estradiol and progesterone) levels using Roche Diagnostic's (Rotkreuz, Switzerland) automated Electrochemiluminescence Immunoassay Elecsys system on a Cobas e411 analyser to stage serum and endometrial samples (Table 3). The assay detection ranges were; 18.4–11010 pmol/L for estradiol (Roche, 06656021190), 0.159–191 nmol/L for progesterone (Roche, 07092539190). The average inter-assay CVs were 7.1% and 13%, respectively.

**Liquid chromatography tandem mass spectrometry (LC-MS/MS)**
**Sample preparation.** Blood serum samples (200 μL) were enriched with isotopically labelled internal standards and transferred to a Supported Liquid Extraction 96-well plate (ISOLUTE, SLE + 400, Biotage, Uppsala, Sweden) along with a calibration curve of multiple steroid standards (0.0025–100 ng). Samples and standards were eluted with 98:2 (v/v) dichloromethane/isopropanol, the resulting extracts reduced to dryness and then reconstituted in 100 μL 70:30 (v/v) water/methanol.

Endometrial tissue (5–10 mg) from a subset of participants with sufficient tissue ($n = 17$ controls, $n = 14$ Long COVID) was placed in 2 mL microfuge tubes containing (2.8 mm) ceramic beads (ThermoFisher Scientific, Leicestershire, UK), suspended in 500 μL 99.9:0.1% v/v acetonitrile/formic acid, enriched with internal standards (20 ng of Working Internal Standard, WIS). Samples were homogenised on a cryo-cooled Bead Mill Homogeniser (Omni International, UK), centrifuged and transferred to an ISOLUTE phospholipid depletion 96-well plate (PLD+, Biotage) alongside a calibration curve of standards and extracted. The final sample numbers for endometrial tissue LC-MS/MS analysis were lower than those in Fig. 2B due to endometrial tissue sample depletion. Menstrual phase: long COVID $n = 3$, control $n = 5$. Proliferative phase: long COVID $n = 7$, control $n = 6$. Secretory phase, long COVID $n = 4$, control $n = 6$.

**Steroid profiling by LC-MS/MS**
Serum and tissue steroid extracts were profiled on an Acquity I-Class UPLC system and QTrap 6500+ mass spectrometer (AB Sciex, UK) system adapted from a previously described LC-MS/MS method[72,73]. All steroid analytical standards used were certified reference material solutions as provided by Sigma-Aldrich/Cerilliant at either 1 mg/mL or 100 mg/mL in methanol or acetonitrile. 20 μL of each calibration standard and sample were sequentially injected onto a Kinetex C18 (150 × 2.1 mm, 2.6 μm) column with a mobile phase system of A-water and 0.05 mM ammonium fluoride, B-methanol and 0.05 mM ammonium fluoride, starting at 50% B, rising to 75% B over 9 minutes, then

100% B by 12 minutes and returning to 50% B at 16 minutes. The eluate from the LC column was transferred to a QTrap 6500+ mass spectrometer instrument for mass analysis by multiple reaction monitoring. The peak area of the steroid, divided by the internal standard, was used to generate calibration curves. Limits of Quantitation (LOQ) for each steroid in serum were 0.05 ng/mL for cortisol, 0.05 ng/mL for cortisone, 0.05 ng/mL for testosterone, 0.01 ng/mL for 5α-dihydrotestosterone, 0.125 ng/mL for progesterone, 0.25 ng/mL for 17α-hydroxyprogesterone, 0.0125 ng/mL for 17beta-estradiol and 0.03 ng/mL for estrone. LOQ for steroids in tissue in ng/g; 0.005 ng for cortisol, testosterone and cortisone, 0.0125 ng for 5α-dihydrotestosterone and 17β-estradiol and 0.125 ng for progesterone and 0.0063 ng for estrone.

**Multiplex automated ELISA**
Blood serum samples were measured for a panel of inflammatory cytokines (IFNG, IL-2, IL-6, IL-8, IL-10 and TNF) using the Ella automated enzyme-linked immunosorbent assay (ELISA) platform (Bio-Techne, Minneapolis, USA), which utilises microfluidic channels to measure each sample in triplicate. Samples were diluted 1:2 and loaded onto SimplePlex Cartridge's (ST01E-PS-005711, Lot# 31895 and 34601), run on an Ella 600-100 system and analysed using SimplePlex software (all Bio-Techne) as per manufacturer's instructions. The assay ranges were; IFNG 0.17–2000 pg/mL, IL-6 0.28–2652 pg/mlL IL-8 0.19–1804 pg/mL, IL-10 0.58–2212 pg/mL, IL-2 0.54–2,050 pg/mL and TNF 0.7–4000 pg/mL. The serum samples that were lower than the SimplePlex assay lot# defined lower limits of quantification (LLOQ), were assigned the lower detection limit value for each assay. IL-2 values were below the limit of detection in all but one serum sample; therefore, these data were excluded.

**AMH ELISA**
AMH levels were measured in proliferative phase blood serum samples using Ansh Labs picoAMH ELISA (Texas, USA, AL-124-i, Lot# 101823) as per manufacturer's instructions. The blood serum samples were measured in duplicate, along with six calibrators and two controls supplied with the kit. The results were read on a TECAN LT-4500 Microplate Absorbance Reader (LabTech, Switzerland) and analysed using SoftMaxPro Validation software (Molecular Devices, California, USA). The assay range was 6–980 pg/mL and the intra-assay CV was 4.0% (0.2–9.5%).

**Reverse transcription quantitative PCR (RTqPCR)**
Total RNA was extracted from endometrial biopsy tissue samples using RNeasy Plus Mini Kits (QIAGEN, Venlo, The Netherlands, 74134) and complementary DNA (cDNA) generated using iScript cDNA Synthesis (Bio-Rad Laboratories, California, USA, 1708890). qPCR assays were designed using the National Centre for Biotechnology Information (NCBI, Maryland, USA) Reference Sequence Database (RefSeq) to confirm target gene identity and Roche Applied Science's (Penzberg, Germany) Universal Probe Library (UPL) Assay Design Centre to select primer pairs and probes (Table S10). TaqMan qPCR was carried out using TaqPath ProAmp Mastermix (ThermoFisher, A30866) on an Applied Biosystems (ABI) QuantStudio 5 real-time PCR system (ThermoFisher). Samples and controls were analysed in triplicate with QuantStudio Design and Assay software using the comparative threshold method. One biological replicate was removed from the

assessment of INFG in endometrial tissue due to Ct values being above the threshold for detection (undetectable). Messenger RNA (mRNA) transcripts were normalized relative to the geomean of two appropriate reference genes, ATP5B and SHDA, as determined with the geNorm algorithm (PrimerDesign, Eastleigh, UK). Samples were quantified relative to a positive endometrial control (mixed menstrual stage) sample.

### Immunohistochemistry (IHC)

Formalin-fixed, paraffin-embedded endometrial biopsy tissue samples were cut to 5-μm sections, dewaxed in Histo-Clear (National Diagnostics, Atlanta, USA, NAT1334) and rehydrated through graded alcohol (100, 95, 80, and 70%) and water. IHC was carried out on a Leica Microsystems (Milton Keynes, UK) BOND-III staining robot using the BOND Polymer Refine Detection system (Leica Microsystems, DS9800), a modified 3,3'-Diaminobenzidine tetrahydrochloride hydrate chromogen and a hematoxylin counterstain protocol, with standard BOND reagents. The specific primary antibody and Refine protocols used are detailed in Table S11. Tissue sections were then dehydrated through graded alcohol (70, 80, 95 and 100%), cleared in Histo-clear and mounted with Pertex (CellPath, Newton, UK, SEA-0100-00A). Slides were digitally recorded with the Axioscan Z1 Brightfield Microscope slide-scanner at ×20 resolution and manipulated using Zen Blue Software (both Zeiss, Carl Zeiss Microscopy, Cambridge UK). All good quality menstrual endometrial images are shown in panels, with 3 representative samples selected for proliferative and secretory phases.

IHC quantification was performed using QuPath Version 0.5.0, with the investigator blind to experimental group[74]. Positive cell detection parameters were used to identify and classify stained nucleated cells in endometrial tissue. Detection thresholds were optimised for each antibody using representative training images. The immune cell marker (CD68 and LL37) positive staining threshold was 0.3, with % of positive nucleated cells reported. Sex receptor (AR and PR) staining threshold was 0.1 and intensities were 0.1–0.4 (weak = 1), 0.4–0.7 (medium = 2) and >0.7 (strong = 3). Histoscore was determined by the sum of % positive nucleated cells after multiplication by staining intensity.

### Statistical analysis

Analysis was carried out using GraphPad Prism 10.4.1. For comparison of multiple datasets with two grouping variables (i.e., long COVID versus controls and stage of menstrual cycle), a two-way analysis of variance was used, with Tukey's multiple comparisons test. A value of $P < 0.05$ was considered significant. Subset analysis of paired samples from the same participant (e.g., serum samples from the secretory and menstrual phases) was analysed using Wilcoxon and paired t-tests, depending on data distribution.

### Reporting summary

Further information on research design is available in the Nature Portfolio Reporting Summary linked to this article.

## Data availability

De-identified survey and app data generated in this study (Figs. 1 and 3) have been deposited on GitHub https://github.com/ataquette/Long-COVID-Mens. Source data for Figs. 4 and 5 are provided with this paper. Any additional information required to reanalyse the data reported in this paper is available from the authors upon reasonable request. Source data are provided with this paper.

## Code availability

All original code has been deposited on GitHub https://github.com/ataquette/Long-COVID-Mens.

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

## Acknowledgements

We would like to acknowledge the contribution of all the participants who took part in these studies and the long COVID patient support group for their input, in particular Ms. Dawn Knight, Ms Natalie Rogers, and Ms Lisa Agyen. We thank research nurses Ms. Sharon McPherson, Ms. Priscilla Fernandez, Ms. Laura Edwards and Ms. Catherine Murray for their assistance with participant recruitment. Dr. Kirsten Wilson performed the immunoassays, Dr. Feng Li assisted with the Ella Multiplex ELISA, and Ms. Charlotte Cuffley contributed to immunohistochemistry. Professor Alistair Williams and Dr Alex Moulla provided an expert histological assessment of endometrial samples. Ms. Lucy Chatwin and Mr. Gavin Sinai assisted with the adaptation of the Balance Menopause App to facilitate our prospective study. Funding for this work was provided by Wellcome Fellowship 209589/Z/17/Z (J.A.M.), Royal Society of Edinburgh R47180 (J.A.M.), Wellbeing of Women RTF1103 (M.W.), ESRC grant ES/P000649/1 (G.K.), MRC grants: G1002033 and MR/N022556/1 (Centre for Reproductive Health).

## Author contributions

J.A.M. wrote the manuscript, was responsible for study conception and design, as well as analysis and interpretation of the data. C.W./M.W./C.R./N.Z.M.H./J.P.S. made substantial contributions to laboratory data acquisition and analysis. A.A., G.K., and Z.O. were responsible for the UK survey conception and design. A.A./G.K. led the analysis and inter-pretation of the survey study data. A.A./L.J. led the analysis and inter-pretation of the prospective study data. H.O.D.C./D.A.G. contributed to the biological study design. All authors revised the manuscript and approved the submitted version.

## Competing interests

The authors declare the following non-financial competing interests: H.O.D.C. has received clinical research support for laboratory consum-ables and staff from Bayer AG and provides consultancy advice (all paid to the institution; no personal remuneration) to Bayer AG, PregLem SA, Gedeon Richter, Vifor Pharma UK Ltd, AbbVie Inc. and Myovant Sciences GmbH. H.O.D.C. has received royalties from UpToDate for an article on abnormal uterine bleeding. J.A.M. has provided consultancy advice (with no personal remuneration) to Gedeon Richter and is a program director at Wellcome Leap. None of these CoI relates to the work described in this manuscript. The remaining authors declare no com-peting interests.
