## [Transparent Peer review file · Nature Communications]

The potential bidirectional relationship between long COVID and menstruation

Corresponding Author: Dr Jacqueline Maybin

Version 0:

Reviewer comments:

Reviewer #1

(Remarks to the Author)

In Studies 2 and 3, the authors have either addressed my previous concerns or provided reasonable explanations why certain points could not be addressed. Study 3 is considerably strengthened by the inclusion of a recovered group, albeit that differences in age among the cohorts meant that analysis of ovarian hormones could not be presented. I have no further comments on these sections.

With respect to Study 1, in the initial submission, I had not understood that the “acute COVID” group was, in most cases, reporting on menstrual experiences several months post-recovery. It may therefore be clearer to refer to this as the “recovered” group. The exclusion of the 301 recently infected respondents makes for a cleaner comparison group and ensures that reported changes are not confounded by the acute phase of infection, addressing my earlier concern. However, it seems that the recently infected respondents may still be included in analyses related to menstrual regularity, flow, duration, intermenstrual bleeding, and missed episodes of menstruation. If they have in fact been excluded from these analyses, this should be stated. Otherwise, given the point discussed above, their exclusion would be appropriate in the final analysis.

Given the number of characteristics analysed, a concluding sentence or summary table indicating which outcomes differed significantly between the non-COVID and the two post-COVID groups would be helpful. Similarly, the abstract would benefit from explicitly stating that menstrual frequency and regularity were unchanged. The current phrase “consistent with our survey findings of no significant change in menstrual frequency or regularity in those with Long COVID” is otherwise confusing, as these findings are not previously mentioned in the abstract.

It is somewhat surprising that the authors do not expect to detect differences in the app dataset among the 372 participants with Long COVID, particularly given the strength of associations observed among 1,048 participants in the survey. Replicating the Study 1 findings using a less biased data source would have been reassuring, but authors do acknowledge recruitment and reporting bias as limitations to their approach. However, more care is needed in discussing ways in which they have mitigated against these biases. In particular, the authors argue that the survey is unlikely to enrich for respondents who noticed menstrual changes, since these were mostly noticed in the context of the vaccine rollout, and were widely discussed in the media from the Spring of 2021, only partially overlapping with their survey. However, discussion of menstrual changes associated with Long COVID started in the Long COVID community in 2020, and were even reported in the medical literature by April 2021 (Davis, 2021, eClinicalMedicine – preprinted in April 2021) so it is likely that at least some of the respondents living with Long COVID would have been aware of these.

Reviewer #2

(Remarks to the Author)

The authors have responded well to my questions and comments.

I am particularly happy that they have included samples from fully recovered COVID patients, which strengthens the paper

even if these participants are few, this gives clear potential for future studies. Since data are only suggestive, it is good that the title of the manuscript has been modified.

Taken together I think the authors have improved the manuscript and added an important control group which strengthens the paper. The topic of studying menstrual consequences of COVID infection in women is original and important, and the paper adds to our knowledge.

Thank you for your considered and constructive review of our manuscript. Please find our responses to your comments below.

Reviewer #1 (Remarks to the Author):

With respect to Study 1, in the initial submission, I had not understood that the “acute COVID” group was, in most cases, reporting on menstrual experiences several months post-recovery. It may therefore be clearer to refer to this as the “recovered” group.

Response: Thank you for raising this important point. We have now added ‘previous’ acute COVID to increase clarity (P4, line 138, P5 lines 153, 181, 192, 200 and P6 line 208 & 217). We have also amended the abstract to make this clearer, (P2, line 23).

The exclusion of the 301 recently infected respondents makes for a cleaner comparison group and ensures that reported changes are not confounded by the acute phase of infection, addressing my earlier concern. However, it seems that the recently infected respondents may still be included in analyses related to menstrual regularity, flow, duration, intermenstrual bleeding, and missed episodes of menstruation. If they have in fact been excluded from these analyses, this should be stated. Otherwise, given the point discussed above, their exclusion would be appropriate in the final analysis.

Response: We have now excluded the 301 recently infected respondents from all analysis and have adjusted the abstract (P2) and results (P4, lines 133 to P5 line 212) to reflect this. Figure 1, Table 1, Supplementary Figure 1 and Tables S1-6 have all been amended.

Given the number of characteristics analysed, a concluding sentence or summary table indicating which outcomes differed significantly between the non-COVID and the two post-COVID groups would be helpful.

Response: Now added to P6, lines 214-218.

Similarly, the abstract would benefit from explicitly stating that menstrual frequency and regularity were unchanged. The current phrase “consistent with our survey findings of no significant change in menstrual frequency or regularity in those with Long COVID” is otherwise confusing, as these findings are not previously mentioned in the abstract.

Response: Unfortunately, due to strict word limits, we were unable to add this detail to the abstract. However, we hope the amended abstract is now clearer and have added a summary paragraph to our results section to explicitly state this (P6, line 216).

It is somewhat surprising that the authors do not expect to detect differences in the app dataset among the 372 participants with Long COVID, particularly given the strength of

associations observed among 1,048 participants in the survey. Replicating the Study 1 findings using a less biased data source would have been reassuring, but authors do acknowledge recruitment and reporting bias as limitations to their approach. However, more care is needed in discussing ways in which they have mitigated against these biases. In particular, the authors argue that the survey is unlikely to enrich for respondents who noticed menstrual changes, since these were mostly noticed in the context of the vaccine rollout and were widely discussed in the media from the Spring of 2021, only partially overlapping with their survey. However, discussion of menstrual changes associated with Long COVID started in the Long COVID community in 2020, and were even reported in the medical literature by April 2021 (Davis, 2021, eClinicalMedicine – preprinted in April 2021) so it is likely that at least some of the respondents living with Long COVID would have been aware of these.

Response: Our prospective app-based survey of 93 women with long COVID examined Long COVID symptoms across the menstrual cycle and did not assess menstrual symptoms. Therefore, replication of our on-line survey findings was not the aim of the app-based study and would be impossible with the data collected. We acknowledge the potential for selection bias of our survey and have added this to our discussion (P10, line 436-437).